# Heterochronic transcription factor expression drives cone-dominant retina development in 13-lined ground squirrels

**Kurt Weir[1†], Pin Lyu[2†], Sangeetha Kandoi[1], Roujin An[1], Nicole Pannullo[1], Isabella Palazzo[1], Jared A Tangeman[1], Jun Shi[3], Steven H DeVries[3], Dana K Merriman[4], Jiang Qian[2]\*, Seth Blackshaw[1,2,5,6,7]\***

[1]Solomon H. Snyder Department of Neuroscience, Johns Hopkins University School of Medicine, Baltimore, United States; [2]Department of Ophthalmology, Johns Hopkins University School of Medicine, Baltimore, United States; [3]Department of Ophthalmology, Northwestern University Feinberg School of Medicine, Chicago, United States; [4]Department of Biology, University of Wisconsin Oshkosh, Oshkosh, United States; [5]Department of Neurology, Johns Hopkins University School of Medicine, Baltimore, United States; [6]Institute for Cell Engineering, Johns Hopkins University School of Medicine, Baltimore, United States; [7]Kavli Neuroscience Discovery Institute, Johns Hopkins University School of Medicine, Baltimore, United States

**\*For correspondence:**
qianjiang007@gmail.com (JQ);
sblack@jhmi.edu (SB)

[†]These authors contributed equally to this work

## eLife Assessment

This **important** study investigates why the 13-lined ground squirrel (13LGS) retina is unusually rich in cone photoreceptors, the cells responsible for color and daylight vision. The authors perform deep transcriptomic and epigenetic comparisons between the mouse and the 13-lined ground squirrel (13LGS) to provide **convincing** evidence that identifies mechanisms that drive rod vs cone-rich retina development. Overall, this key question is investigated using an impressive collection of new data, cross-species analysis, and subsequent in vivo experiments.

**Abstract** Evolutionary adaptation to diurnal vision in ground squirrels has led to the development of a cone-dominant retina, in stark contrast to the rod-dominant retinas of most mammals. The molecular mechanisms driving this shift remain largely unexplored. Here, we perform single-cell RNA sequencing and chromatin accessibility profiling (scATAC-Seq) across developmental retinal neurogenesis in the 13-lined ground squirrel (13LGS) to uncover the regulatory basis of this adaptation. We find that 13LGS cone photoreceptors arise not only from early-stage neurogenic progenitors, as seen in rod-dominant species like mice, but also from late-stage neurogenic progenitors. This extended period of cone generation is driven by a heterochronic shift in transcription factor expression, with cone-promoting factors such as *Onecut2*, *Pou2f1*, and *Zic3* remaining active in late-stage progenitors, and factors that promote cone differentiation such as *Thrb*, *Rxrg*, and *Mef2c* expressed precociously in late-stage neurogenic progenitors. Functional analyses reveal that *Zic3* and *Mef2c* are sufficient to promote cone and repress rod photoreceptor-specific gene expression and act through species-specific regulatory elements that drive their expression in late-stage progenitors. These results demonstrate that modifications to gene regulatory networks underlie the development of cone-dominant retinas and provide insight into mechanisms of sensory adaptation and potential strategies for cone photoreceptor regeneration in vision disorders.

## Introduction

Evolutionary adaptation to new ecological niches often requires the development of new sensory, motor, and/or behavioral capacities. These adaptations are typically mediated by structural changes to the nervous system which arise from changes in the composition and connectivity of individual neuronal subtypes (*Arendt, 2008*; *Katz, 2011*). While the molecular mechanisms underlying these changes are generally poorly understood, the vertebrate retina serves as an ideal system for investigating them. Shifts in temporal niche—such as transitions between predominantly nocturnal and diurnal behavior—have occurred repeatedly across diverse vertebrate lineages (*Le Duc and Schöneberg, 2016*; *Slavenko et al., 2022*; *Walls, 1942*). These shifts inevitably require adaptive modifications in the rod and cone visual pathways, which detect dim and bright light, respectively.

Mammals represent a notable exception to the general trend in which diurnal vertebrate species possess cone-dominant retinas. This deviation reflects the long-term effects of a selective nocturnal bottleneck that occurred before the Cretaceous–Paleogene mass extinction event, preceding the mammalian radiation (*Walls, 1942*; *Gerkema et al., 2013*). Despite considerable variation in photoreceptor subtype composition among mammals (*Ahnelt and Kolb, 2000*; *Peichl, 2005*; *Kaskan et al., 2005*), even strongly diurnal species typically have rod-dominant retinas, with rod-to-cone ratios ranging from approximately 5:1 to 20:1 (*Peichl, 2005*; *Mustafi et al., 2009*). Strictly nocturnal species, such as mice, exhibit even more pronounced rod dominance of 33:1 (*Jeon et al., 1998*). However, diurnal ground squirrels are a major exception to this rule, possessing an inverted rod-to-cone ratio, having only one rod for every 6–7 cones throughout most of the retina (*Verra et al., 2020*; *Kryger et al., 1998*), and as many as 50 cones for every rod in the central visual streak (*Kryger et al., 1998*). By examining the development of the cone-dominant ground squirrel retina and comparing it to that of nocturnal rodents like mice, we can identify molecular mechanisms underlying this dramatic and evolutionarily rapid shift in photoreceptor subtype composition.

In rod-dominant mouse and human retinas, cones are generated primarily during early stages of retinal neurogenesis, whereas rods are predominantly produced in later stages (*Zhang et al., 2023*; *Clark et al., 2019*; *Cepko, 2014*; *Lyu et al., 2021*). The progenitors that selectively give rise to cones and rods are in turn distinguished by the activity of a network of transcription factors (TFs) that confer early and late-stage temporal identity. The progenitors that give rise to cones and rods are distinguished by the activity of a network of TFs that establish early- and late-stage temporal identities. Cones originate from a subpopulation of early-stage *Otx2*- and *Neurod1*-positive neurogenic progenitors that express *Onecut1/2* and *Pou2f1/2*, which confer competence to generate both cones and horizontal cells and activate TFs that promote cone differentiation while simultaneously repressing rod-specific factor (*Emerson et al., 2013*; *Javed et al., 2020*). In contrast, late-stage *Otx2*- and *Neurod1*-positive neurogenic progenitors primarily generate postmitotic rod precursors, which express TFs such as *Nrl* and *Nr2e3* that suppress cone-specific genes, while also activating rod-specific genes (*Chen et al., 2005*; *Milam et al., 2002*; *Cheng et al., 2004*; *Mears et al., 2001*; *Daniele et al., 2005*). Postmitotic cone precursors, in turn, express TFs such as *Thrb*, *Rxrg*, and *Sall3*, which further activate cone-specific genes and inhibit rod-specific gene expression (*Swaroop et al., 2010*; *de Melo et al., 2011*). Evolutionary changes in the rod-to-cone ratio may result directly from alterations in the expression or activity of these factors, or from other, as-yet-uncharacterized components of the transcriptional regulatory networks that govern temporal patterning and photoreceptor identity.

To investigate the molecular mechanisms underlying the development of cone-dominant mammalian retinas, we performed unbiased single-cell RNA sequencing (scRNA-Seq) and ATAC-Seq analyses to comprehensively profile gene expression and regulatory changes in the developing retina of the cone-dominant 13-lined ground squirrel (13LGS) across the full course of neurogenesis. We directly compared these data with similar datasets from the developing retina of rod-dominant mice. Our findings reveal that cone generation occurs at high levels even during late neurogenesis in 13LGS, coinciding with two broad heterochronic shifts in TF expression in late-stage retinal progenitors. The first shift involves an expanded expression of a subset of TFs that are typically active in early-stage progenitors in rod-dominant retinas—factors known or predicted to promote cone specification. In addition to previously characterized factors such as *Onecut2* and *Pou2f1*, this group includes previously uncharacterized genes, such as *Zic3*, which we show to be both necessary and sufficient for cone specification during early neurogenesis. The second shift involves precocious expression of TFs typically found in postmitotic cone precursors, including *Thrb* and *Rxrg*, along with newly identified factors

such as *Mef2c*, which cooperatively activate cone-specific genes while repressing rod-specific ones. The expression of these TFs in both late-stage progenitors and committed photoreceptor precursors is driven by multiple 13LGS-specific enhancer elements. These enhancers are directly targeted by *Otx2* and *Neurod1*, which broadly regulate photoreceptor specification and differentiation, as well as by heterochronically expressed TFs specific to early-stage retinal progenitors and cone precursors. Our results demonstrate that heterochronic expansion of the expression of TFs that promote cone development is a key event in the development of the cone-dominant 13LGS retina.

## Results

### Single-cell RNA- and ATAC-Seq profiling of developing 13LGS retina

In 13LGS, retinal neurogenesis begins around embryonic day 18 (E18), with birth occurring at E27 (blue-dotted line in *Figure 1A*), and concludes by postnatal day 17 (P17) (*Mossman and Weisfeldt, 1939*). To comprehensively profile cellular-level changes in gene expression and chromatin accessibility, we conducted scRNA- and scATAC-Seq analyses across ten developmental time points, spanning the full course of neurogenesis and the immediate post-neurogenesis period (E18–P21). We selected these time points to align with stages previously profiled in the developing mouse retina (E11–P14) (*Figure 1A*; *Lyu et al., 2021*; *Clark et al., 2019*). High-quality data were obtained for all stages, with scATAC-Seq data showing expected fragment length and distribution patterns (*Figure 1—figure supplement 1A, B*). Using selective markers of cell types previously identified in the developing mouse retina, we successfully annotated cell populations identified through UMAP clustering of aggregated scRNA- and scATAC-Seq data. This approach distinguished all major retinal cell types, including both early- and late-stage primary and neurogenic retinal progenitor cells (RPCs and N. RPCs) (*Figure 1B*).

Cell type-specific expression and chromatin accessibility patterns showed greatest correlation in the same and closely related cell types across modalities (*Figure 1—figure supplement 1C*), with expected expression and accessibility patterns observed for known cell type-specific marker genes (*Figure 1C*). Each annotated progenitor state and cell type (C1–C12) exhibited significant numbers of differentially expressed genes (DEGs) (*Figure 1—figure supplement 1D*, *Supplementary file 1*) and differentially accessible chromatin regions (DARs) (*Figure 1—figure supplement 1E*). We investigated enrichment of monomeric TF motifs within DARs. Conserved patterns of motif accessibility, identified using ChromVAR and the TRANSFAC2018 database (*Figure 1—figure supplement 1F*, *Supplementary file 1*) and TF footprinting (*Figure 1—figure supplement 1G, H*) were observed. Notably, motifs annotated as belonging to *Nr2f1* showed strong activity in early-stage RPCs, *Pou4f2* in retinal ganglion cells, *Nfix* in late-stage RPCs and Müller glia, and *Crx* and *Neurod1* in photoreceptor precursors (*Figure 1—figure supplement 1F*, *Supplementary file 1*). Motifs assigned to one TF can have more complicated binding situations in vivo, but these enrichments are markedly consistent with previous studies in developing mouse and human retina (*Lyu et al., 2021*; *Thomas et al., 2022*).

Direct comparisons of cell type composition across individual time points in the 13LGS and mouse retina indicated a high degree of stage-matching and broad concurrence (*Figure 1—figure supplement 1I*). The transition from early- to late-stage states occurred between E26 and P1 in 13LGS, compared to E16–E18 in mice (*Lyu et al., 2021*; *Clark et al., 2019*). The expected temporal patterns of neurogenesis were observed in both species: retinal ganglion cells and horizontal cells were generated predominantly in the early stage, amacrine cells in the late embryonic and early postnatal period, and bipolar cells and Müller glia were produced in the late stage (*Supplementary file 1*). Photoreceptor precursors were detected from E18 through P12 in 13LGS, with cone cells becoming clearly distinct in the early postnatal period. Notably, a large proportion of cones were generated during the first postnatal week in 13LGS. As expected, identification of mature cones in the 13LGS scATAC data preceded the scRNA data, as chromatin opening precedes gene expression (*Ma et al., 2020*). In contrast to mice, where rods are distinct from E18 onward (*Clark et al., 2019*), rods in 13LGS were not detected until the late postnatal period (P12).

Both scRNA- and scATAC-Seq data demonstrated that, at all developmental stages, cones were significantly more abundant than rods in 13LGS, comprising approximately 25% of all retinal cells following the completion of neurogenesis at P12 (*Supplementary file 1*). By subclustering cones, we can separate distinct clusters of S-cones from the far more numerous M-cones at each postnatal time

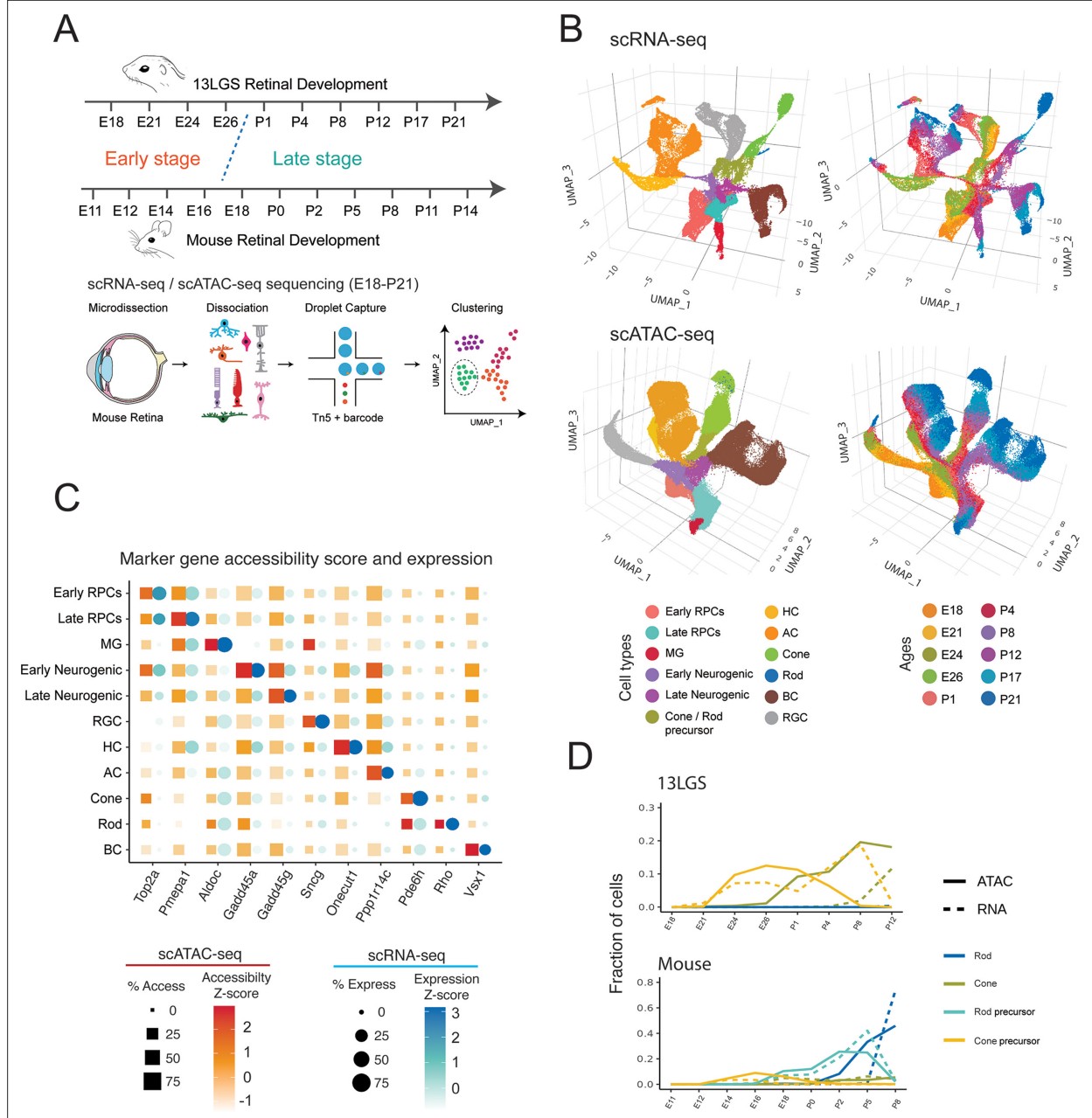

**Figure 1.** Overview of the study. (**A**) Schematic summary of the study. scRNA- and scATAC-Seq of whole 13LGS retinas were performed at 10 different time points. Major retinal cell types are identified by clustering in both scRNA- and scATAC-Seq datasets. (**B**) UMAP projections of all 13LGS retinal cells from scRNA- and scATAC-Seq. Each point represents a single cell, with cell type and time point indicated by shading. (**C**) Cell-specific marker gene expression and accessibility are shown across 11 major retinal cell types. (**D**) The relative abundances of rods, rod precursors, cones, and cone precursors are different between developing 13LGS and mouse retina.

The online version of this article includes the following figure supplement(s) for figure 1:

**Figure supplement 1.** Quality control of scATAC- and scRNA-Seq data.

point, except P21 where there are too few S-cones present to form a separate cluster, and identify genes enriched in each cone subtype (***Supplementary file 2***; ***Kryger et al., 1998***). In contrast, in mice, rods progressively increased in proportion, reaching approximately 70% of retinal cells by P8 (***Figure 1D***). These proportions are roughly in agreement with what has been found in vivo by prior studies using orthogonal approaches (***Jeon et al., 1998***; ***Kryger et al., 1998***).

# Comparative multiomic analysis of photoreceptor differentiation in 13LGS and mouse

These findings suggest that late-stage RPCs in 13LGS retina are competent to generate cone photoreceptors but not other early-born cell types, with rod specification occurring only at very late postnatal stages (>P12) of neurogenesis. To investigate the mechanisms underlying this heterochronic shift in developmental competence, we used Harmony on conserved genes to integrate scRNA- and scATAC-Seq data from all late-stage time points in 13LGS and mice (*Figure 2A*, *Supplementary file 3*).

In the 13LGS scRNA-Seq data, we observed well-defined, continuous differentiation trajectories linking late-stage primary and neurogenic RPCs to cones, bipolar, and amacrine cells. However, no clear trajectory for rod differentiation was detected, likely due to the very low number of rod cells detected prior to P17 (*Figure 2A*). In contrast, scRNA-Seq data from mice revealed differentiation trajectories connecting neurogenic progenitors to rods, amacrine, and bipolar cells, but not to cones at this stage (indicated by arrows in *Figure 2A*). Similarly, scATAC-Seq analysis in 13LGS identified a strong progenitor-to-cone photoreceptor trajectory, but no corresponding progenitor-to-rod trajectory (*Figure 2B*). Interestingly, both species exhibited a bipolar cell/photoreceptor precursor population (*Figure 2A, B*). This may suggest that late-stage 13LGS retina contains bipolar cell/cone bipotent progenitors similar to the bipolar cell/rod bipotent progenitors of late-stage mouse retina (*Lyu et al., 2021*; *Clark et al., 2019*), though further investigation would be required to confirm this. Nonetheless, this highlights the specificity of the effects of heterochronic expansion of cone-promoting factors.

To further characterize species-specific patterns of gene expression and regulation during postnatal photoreceptor development, we analyzed differential gene expression, chromatin accessibility, and motif enrichment across late-stage primary and neurogenic progenitors, immature photoreceptor precursors, rods, and cones. Due to their very low number before time point P17, we were unable to include 13LGS rods in the analysis. This analysis revealed eight distinct species-specific differential expression patterns identified by *k*-means clustering (C1–C8) (*Figure 2C, D*; *Supplementary file 4*).

Clusters C1–C3 were selectively expressed in 13LGS. C1 was active exclusively in RPCs, C2 active across all stages of cone specification, and C3 restricted to mature cones. Notably, C2 included several genes previously implicated in photoreceptor development, such as the cone-promoting TF *Onecut1* (*Emerson et al., 2013*; *Sapkota et al., 2014*) and *Mef2c*, a MADS-box TF which is cone-enriched in human retina (*Wolf et al., 2017*; *Kallman et al., 2020*) and reported to be essential for photoreceptor survival (*Nagar et al., 2017*). The smaller cluster C3 contained TFs such as *Thrb*, *Rxrg*, and *Sall3*, which are all selectively expressed in cone precursors and essential for cone-specific gene expression (*de Melo et al., 2011*; *Roberts et al., 2005*; *Ng et al., 2001*). Additionally, *Zic3*, which is selectively expressed in early-stage RPCs in mice (*Lyu et al., 2021*; *Clark et al., 2019*), was also enriched in C3.

Clusters C4–C8 contained genes that were selectively active in mice relative to 13LGS (*Figure 2C*). Clusters C4–C6 showed higher expression levels in mice across different cellular contexts: C4 was specific to progenitors, C5 was active throughout specification, and C6 spanned progenitors and photoreceptor precursors. C4 included the early-stage RPC TF *Nr2f1*, C5 contained *Sox8* and *Sox11*, and C6 featured *Dlx2*. While the direct relevance of most of these differences remains unclear, *Nr2f1* is essential for establishing the dorsal–ventral gradients in M- and S-cone opsin expression that are observed in mouse cones (*Satoh et al., 2009*), but which do not exist in 13LGS (*Kryger et al., 1998*).

Clusters C7 and C8, which were selectively active in mouse photoreceptor precursors and rods, contained several TFs known to promote rod differentiation and repress cone fate. In C7, we identified *Casz1*, a temporal patterning factor that represses the competence of late-stage progenitors to generate cones, while also promoting rod specification (*Mattar et al., 2015*). In C8, as expected, we observed *Nrl*, *Nr2e3*, and *Samd7*, all of which activate rod-specific genes while repressing cone-specific genes (*Chen et al., 2005*; *Cheng et al., 2004*; *Hao et al., 2012*; *Omori et al., 2017*).

Integrated scRNA- and ATAC-Seq analysis using ArchR identified regulatory elements predicted to control each of these genes in both 13LGS and mouse (*Figure 2D*). Notably, genes in C2 and C3, which were more highly expressed in 13LGS and are strong candidates for promoting cone specification, exhibited significantly higher numbers of predicted functional *cis*-regulatory elements than their mouse counterparts (*Figure 2E*). This species-specific increase in predicted regulatory elements was either absent from, or far less pronounced in, other gene clusters.

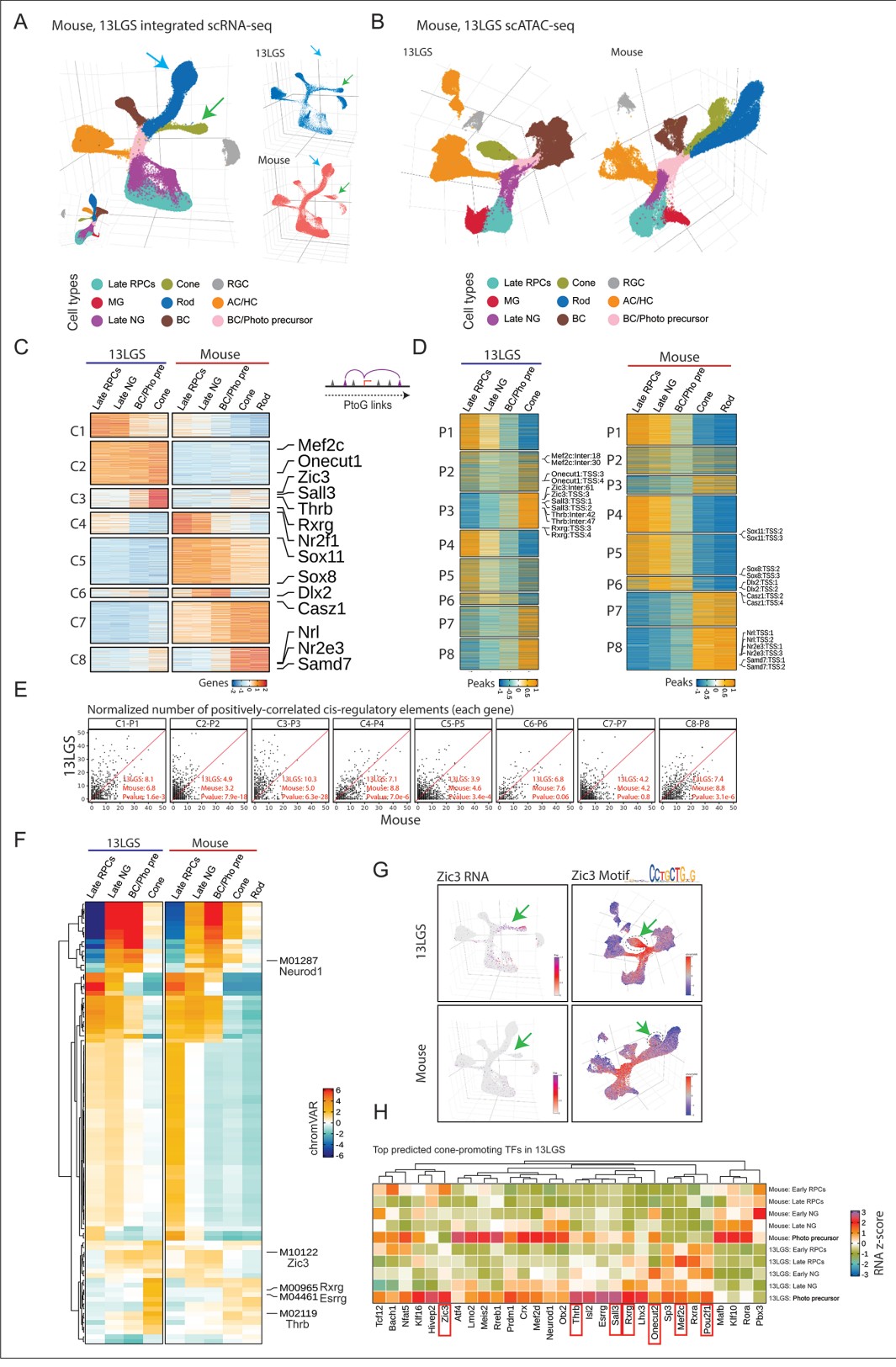

**Figure 2.** Identification of 13LGS-specific regulatory mechanisms in cone development by integrating late-stage retinal scRNA- and scATAC-Seq data with mouse. (**A**) UMAP projections of the integrated scRNA-Seq data from 13LGS and mouse. (**B**) UMAP projections of the scATAC-Seq data, showing cell types re-annotated based on the integrated scRNA-Seq analysis. (**C**) Heatmaps displaying eight clusters of genes that are differentially expressed

*Figure 2 continued*

between 13LGS and mouse. Column labels correspond to the five major cell types relevant to photoreceptor development: Late RPCs (late-stage primary retinal progenitor cells), Late NG (late-stage neurogenic progenitor cells), BC/Pho pre (bipolar cell, cone, and rod precursor cells), cone photoreceptors, and rod photoreceptors. (**D**) Heatmaps showing the chromatin accessibility of positively correlated regulatory elements for each of the eight differentially expressed gene (DEG) clusters. (**E**) Dot plots depicting the relative number of positively correlated regulatory elements for each conserved gene pair between 13LGS and mouse. Each dot represents a conserved gene pair, with the *x*-axis indicating the number of regulatory elements in mouse and the *y*-axis showing the number in 13LGS. Pairwise *t*-tests were used to compare the number of elements in each species for each gene pair, and the median values are indicated below each plot. (**F**) Differential motif analysis in five cone development-associated cell types between 13LGS and mouse. (**G**) *Zic3* expression motif accessibility is higher in 13LGS photoreceptor precursors and mature cones than in mouse. (**H**) Heatmap showing the expression levels of the top 30 cone-promoting genes identified by gene regulatory network (GRN) analysis in 13LGS across retinal progenitor cells (RPCs), neurogenic progenitor cells (N. RPCs), and photoreceptor precursors in both 13LGS and mouse.

The online version of this article includes the following figure supplement(s) for figure 2:

**Figure supplement 1.** Transcription factors that show heterochronic gene expression and co-expression with Otx2 show sustained expression in 13LGS retinas.

## Identifying TFs that promote cone specification in 13LGS

To identify TFs that strongly contribute to cone specification, we compared differential motif enrichment in accessible chromatin regions across cell types between 13LGS and mice (*Figure 2F*, *Supplementary file 4*). Motifs associated with neurogenic bHLH factors such as Neurog2, Ascl1, Olig2, and Neurod1 were highly enriched in neurogenic RPCs and photoreceptor precursors in both species, as was the motif for Otx2, a key factor required for both rod and cone specification (*Nishida et al., 2003*; *Emerson et al., 2013*). Similarly, motifs for TFs strongly expressed in late-stage RPCs, such as Lhx2 and Sox8, were prominent in progenitors of both species.

Motifs for cone-specific TFs, including Thrb, Rxrg, and Esrrg, were significantly more prominent in cones of 13LGS. Furthermore, in 13LGS, the Zic3 motif was accessible in both progenitors and mature cones, whereas in mice, its accessibility declined in mature cones (*Figure 2F, G*). These data suggest that species-specific gene regulatory networks (GRNs) observed in 13LGS late-stage RPCs are characterized by relatively higher activity levels of TFs that are enriched in both early-stage RPCs and postmitotic cone precursors in mice and contribute to the development of the cone-dominant retina in 13LGS.

To more systematically identify TFs regulating cone specification in 13LGS, we used scRNA- and ATAC-Seq data from late-stage RPCs and differentiating photoreceptors to reconstruct GRNs predicted to activate cone-specific genes (*Figure 2H*, *Supplementary file 5*). This analysis focused on TFs with differential expression in late-stage RPCs between 13LGS and mice. Some TFs in this network, such as *Otx2*, *Neurod1*, *Crx*, *Prdm1*, and *Mef2d*, have been shown to promote both rod and cone differentiation (*Furukawa et al., 1999*; *Nishida et al., 2003*; *Pennesi et al., 2003*; *Katoh et al., 2010*; *Andzelm et al., 2015*), and their expression patterns did not exhibit strong species-specific differences.

In contrast, two broad patterns of differential expression of cone-promoting TFs were observed between mouse and 13LGS. First, TFs identified in this network that are known to be required for committed cone precursor differentiation, including *Thrb*, *Rxrg*, and *Sall3* (*Swaroop et al., 2010*; *de Melo et al., 2011*; *Lu et al., 2020*), consistently showed stronger expression in both late-stage RPCs and early-stage primary and/or neurogenic RPCs of 13LGS compared to mice. Other genes such as *Mef2c*, which, while not functionally linked to cone specification, are also most prominently expressed in differentiated cones, also showed expression in late-stage RPCs. This suggests a species-specific pattern of precocious expression of cone differentiation-promoting TFs in the course of cone differentiation. Second, TFs in the network known to promote cone specification in early-stage mouse RPCs, such as *Onecut2* and *Pou2f1*, exhibited enriched expression in early and late-stage primary and/or neurogenic RPCs of 13LGS, implying a heterochronic expansion of cone-promoting factors into later developmental stages. This pattern was also observed for *Zic3*, which emerged as the most likely candidate for driving cone-specific gene expression in 13LGS. In mice, *Zic3* was selectively enriched in early-stage RPCs (*Lyu et al., 2021*; *Clark et al., 2019*), supporting its potential role in

cone specification. In contrast, genes such as *Casz1*, which act in late neurogenic RPCs to promote rod specification (*Mattar et al., 2015*), are downregulated in 13LGS late neurogenic RPCs relative to mice.

To confirm heterochronic expression of candidate cone-promoting factors in late-stage developing 13LGS retina, we performed immunohistochemistry (IHC) and RNA-hybridization chain reaction (HCR) analysis. We analyzed co-expression of the candidate cone-promoting factors *Zic3* and *Mef2c*, as well as *Pou2f1*, a cone-promoting factor active in early-stage RPCs in mouse (*Emerson et al., 2013*; *Javed et al., 2020*), with markers of neurogenic RPCs and/or differentiating cones. *Pou2f1* co-expression with *Otx2* was observed at P1, P5, P10, and P24 in 13LGS (*Figure 2—figure supplement 1A*). At all ages examined, *Zic3* was co-expressed with the neurogenic RPC/photoreceptor precursor marker *Otx2* and the cone precursor marker *Rxrg* (*Figure 2—figure supplement 1B-E*).

Weak Mef2c protein expression was observed in Otx2-positive neurogenic RPCs at P1 (data not shown), with increased expression at P5, with stronger expression detected in mature photoreceptors at later developmental stages (*Figure 2—figure supplement 1F-G*).

## *Zic3* promotes cone-specific gene expression and is necessary for generating the full complement of cone photoreceptors

Previous studies have demonstrated that overexpression of either *Onecut1* or *Pou2f1* in late-stage mouse retinal progenitors is sufficient to induce the expression of cone-specific genes while also repressing rod-specific genes (*Javed et al., 2020*; *Emerson et al., 2013*). We employed a similar approach to determine whether *ZIC3*, either alone or in combination with the heterochronically expressed cone-promoting factor *POU2F1*, can likewise induce cone differentiation from late-stage mouse RPCs. To test this, we conducted electroporation of P0 ex vivo mouse retinal explants, as previously described (*Melo and Blackshaw, 2018*), and used scRNA-Seq, running different conditions together using CellPlex, to assess the effects of *ZIC3* overexpression alone, or in combination with *POU2F1* (*Figure 3A*). ScRNA-Seq analysis of GFP-positive electroporated cells at 5 days in vitro showed that overexpression of heterochronically expressed TFs produced a cone-like photoreceptor precursor population (*Supplementary file 6*, *Figure 3—figure supplement 1A*). Comparison of gene expression between this photoreceptor cluster and rod photoreceptors revealed that empty plasmids induced one DEG: *GFP* (*Figure 3B, C*, *Supplementary file 6*). *ZIC3* not only induced the expression of several cone photoreceptor-specific genes (*Uhlén et al., 2015*) at this stage, it also downregulated the expression of many rod-specific genes (*Figure 3B, C*). *POU2F1* overexpression upregulated an overlapping but distinct and larger set of cone-specific genes relative to *ZIC3*, while also downregulating many of the same rod-specific genes, often to a greater extent (*Figure 3C*). Co-overexpression of *ZIC3* and *POU2F1* in late-stage RPCs had a synergistic effect, resulting in more DEGs than did either *ZIC3* or *POU2F1* alone, and greater quantitative changes in cellular levels of expression (*Figure 3B, C*, *Supplementary file 6*).

*Onecut1* overexpression similarly produced a population of cone-like precursors (*Figure 2—figure supplement 1B*, *Supplementary file 6*). It upregulated a largely distinct set of cone-specific genes compared to *POU2F1* alone but shared some upregulated cone-specific genes with the combined overexpression of *ZIC3* and *POU2F1* (*Figure 2—figure supplement 1C, D*, *Supplementary file 6*). Furthermore, *Onecut 1* overexpression downregulated many of the same rod-specific genes as did the combined overexpression of *POU2F1* and *ZIC3* (*Figure 2—figure supplement 1C, D*). Similarly to *Onecut 1* and *POU2F1* overexpression, *ZIC3* overexpression produced horizontal cell-like precursors in addition to cone-like precursors (*Javed et al., 2020*; *Emerson et al., 2013*; *Figure 3—figure supplement 1A, B*, *Supplementary file 6*).

Because conventional *Zic3* knockouts show widespread and complex developmental defects and high gestational lethality (*Purandare et al., 2002*), we generated an early RPC-specific *Zic3* loss-of-function mutant by combining the retinal progenitor-specific Chx10-CreGFP transgenic line with a conditional *Zic3* allele (*Rowan and Cepko, 2004*; *Sutherland et al., 2013*; *Figure 3D*). This resulted in a statistically significant ~20% reduction in the density of cone photoreceptors in the mutant retina (*Figure 3E, F*), while the relative numbers of rods and horizontal cells remained unaffected (*Figure 3—figure supplement 2A-D*).

ScRNA-Seq analysis (*n* = 4) of P14 *Zic3* conditional mutant retinas identified quantitative changes in gene expression in several terminally differentiated cell types (*Figure 3—figure supplement 2E-G*,

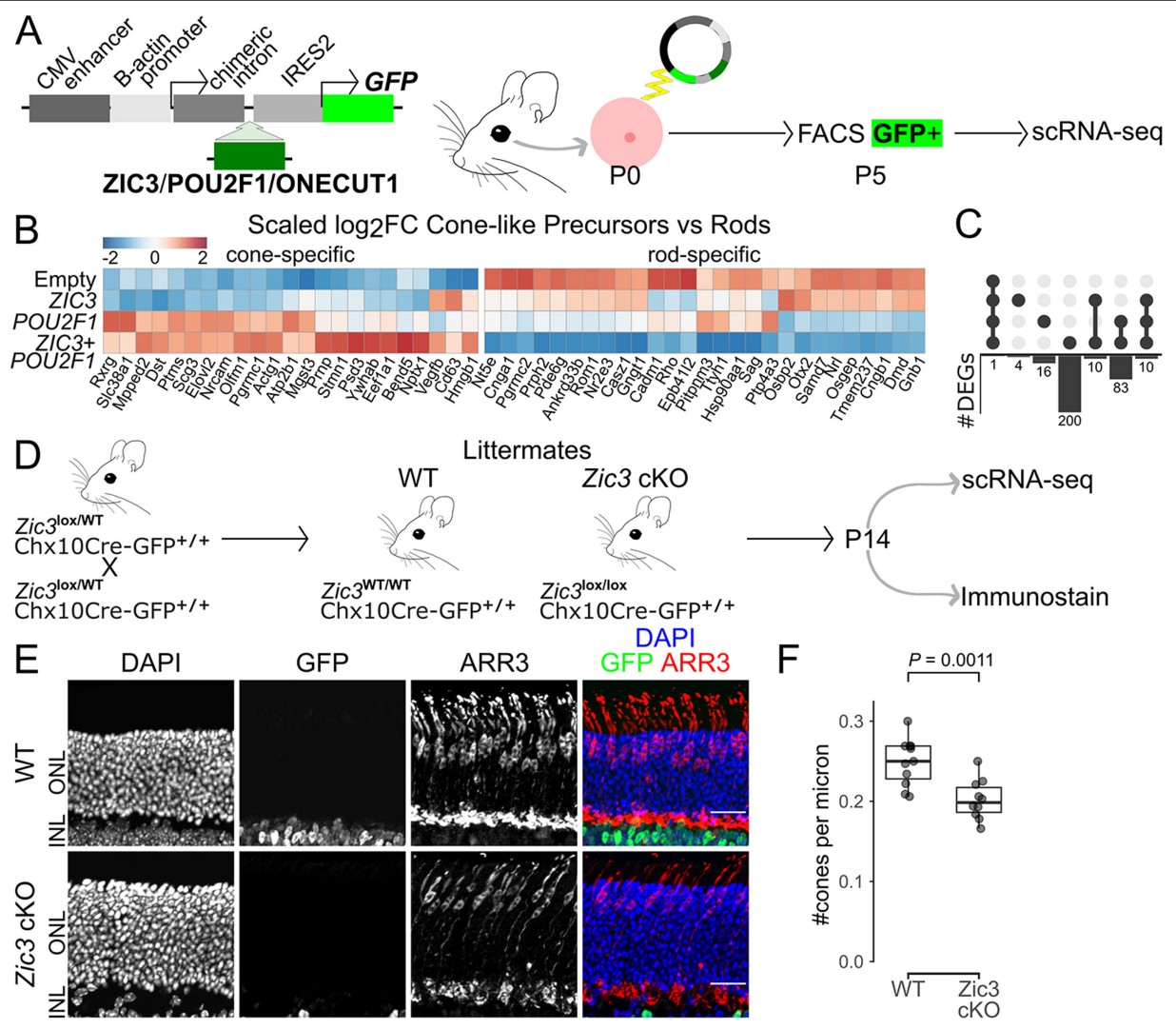

**Figure 3.** Zic3 is necessary for normal mouse cone development and sufficient to promote cone gene expression. (**A**) Diagram of overexpression strategy used to test effects of *ZIC3* overexpression. (**B**) Heatmap of $\log_2$ fold change in expression between cone-like precursors and rods separated by condition for select genes, scaled by gene. (**C**) Upset plot of significant differentially expressed genes shared between conditions. (**D**) Diagram of retinal progenitor cell-specific conditional loss of function analysis of *Zic3*. (**E**) Immunohistochemistry showing GFP and Arr3 expression in P14 wild type (WT) and *Zic3*$^{lox/lox}$ (*Zic3* cKO) mouse retinas. Scale bars, 20 µm. (**F**) Box plot of the number of cones per micron along the retinal slice ($n$ = 11 WT, 10 *Zic3* cKO). p-values calculated by Wilcoxon rank-sum test. P0, postnatal day 0; P5, postnatal day 5; P14, postnatal day 14; FACS, fluorescence-activated cell sorting; scRNA-Seq, single-cell RNA sequencing; ONL, outer nuclear layer; INL, inner nuclear layer; DAPI, 4',6-diamidino-2-phenylindole; GFP, green fluorescent protein; OE, overexpression; $\log_2$FC, $\log_2$ fold change; DEG, differentially expressed gene; WT, wild type; cKO, conditional knockout.

The online version of this article includes the following figure supplement(s) for figure 3:

**Figure supplement 1.** *ZIC3* overexpression promotes the formation of cone-like photoreceptor precursor cells.

**Figure supplement 2.** *Zic3* is required for normal patterns of gene expression in cones and Müller glia.

---

*Supplementary file 7*). The remaining mutant cones showed reduced expression of *Thrb* but increased expression of multiple cone-specific markers, including *Opn1sw*, *Pde6h*, and *Gngt2* (*Figure 3—figure supplement 2F*). Despite the increased *Opn1sw* expression, there does not seem to be a differential impact on genes enriched in S- or M-cones. For instance, *Pde6h*, which we find to show reduced expression in S-cones relative to M-cones in 13LGS, is increased in mutant cones and *Pex5l*, which is higher in S-cones, is reduced in mutant cones (*Figure 3—figure supplement 2F*, *Supplementary file 7*). In addition, we observe a broad decrease in expression of genes expressed at high levels in both cones and rods (*Rpgrip1*, *Drd4*) and rod-specific genes (*Rho*, *Cnga1*, *Pde6b*) in mutant cones

(*Figure 3—figure supplement 2F*). Since rods are fragile cells that are located immediately adjacent to cones, some level of contamination of rod-specific genes is inevitable in single-cell analysis of dissociated cones (*Clark et al., 2019*; *Lyu et al., 2021*), and this reduced level of rod contamination could result from altered adhesion between mutant rods and cones. In contrast, increased expression of rod-specific genes (*Rho*, *Nrl*, *Pde6g*, *Gngt1*) and pan-photoreceptor genes (*Crx*, *Stx3*, *Rcvrn*) was observed in Müller glia (*Figure 3—figure supplement 2G*), which may likewise result from altered adhesion between Müller glia and rods. Finally, several Müller glia-specific genes were downregulated, including *Clu*, *Aqp4*, and Notch pathway components such as *Hes1* and *Id3*, with the exception of *Hopx*, which was upregulated (*Figure 3—figure supplement 2G*). This likely reflects the indirect effects of *Zic3* loss of function in retinal progenitors. These findings indicate that *Zic3* is essential for the proper expression of photoreceptor genes in cones while also playing a role in regulating expression of Müller glia-specific genes.

## *MEF2C* overexpression promotes cone-specific gene expression and inhibits rod photoreceptor maturation

We next investigated the effects of both *MEF2C* overexpression in late-stage RPCs and RPC-specific loss of function of *Mef2c* in mice. As described for *ZIC3*, we conducted ex vivo electroporation of P0 mouse retinal explants with plasmid vectors expressing *MEF2C*, using empty vectors that expressed GFP alone as controls (*Figure 4A*). Since *Mef2c* expression is initiated at later stages of cone development relative to *Zic3* (*Figure 3A*, *Figure 2—figure supplement 1A*), explants were cultured until P8 for both scRNA-Seq and immunohistochemical analysis.

ScRNA-Seq analysis revealed substantial alterations in cell type composition following *MEF2C* overexpression, with a marked increase in both neurogenic RPCs and immature photoreceptor precursors, which suggested a broad delay or arrest in rod differentiation (*Figure 4B, C*, *Supplementary file 8*). Additionally, a novel subpopulation of cone-like precursors emerged (*Figure 4B*). Cone-like precursors exhibited upregulation of genes such as *Gnat2*, *Arr3*, and *Pde6c*, while also maintaining expression of genes that are typically enriched in both immature rod and cone photoreceptors, including *Tpi1*, *Rtn4*, and *Tpt1* (*Figure 4C*).

Histological analysis indicated that *MEF2C* overexpression consistently led to morphological abnormalities in electroporated regions, including the formation of whorls and rosettes in the outer nuclear layer, a phenotype reminiscent of *Nrl* and *Nr2e3* mutants that exhibit ectopic cone-specific gene expression in rods (*Figure 4D*, *Figure 4—figure supplement 1A-B*, *Mears et al., 2001*; *Akhmedov et al., 2000*). Although a few GFP-positive electroporated cells co-expressing the cone-specific marker *Gnat2* were detected in control (likely due to the electroporation of cone precursors, which we have previously observed in P0 retinal explants; *Onishi et al., 2010*; *Clark et al., 2019*; *Lyu et al., 2021*; *Leavey et al., 2025*), there was a significant increase in double-positive cells in the test condition, matching the novel cone-like precursor population found in the scRNA-Seq dataset (*Figure 4E*). No obvious reduction in the relative number of Nrl-positive cells was observed (*Figure 4—figure supplement 1A*). No changes were observed in the distribution or abundance of other retinal cell types (*Figure 4—figure supplement 1B-D*).

A previous study reported that *Nestin-Cre;Mef2c^lox/lox* mice, which are predicted to exhibit loss of function of *Mef2c* throughout the developing central nervous system, showed a rapid degeneration of both rod and cone photoreceptors that resembled photoreceptor loss observed in fast-acting *rd1/rd1* mutants (*Nagar et al., 2017*). To evaluate the effects of retinal progenitor-specific loss of function of *Mef2c*, we generated *Chx10-Cre;Mef2c^lox/lox* mice (*Vong et al., 2005*; *Rowan and Cepko, 2004*; *Figure 4—figure supplement 2A*). No changes in cone numbers were observed at P14 (*Figure 4—figure supplement 2B, C*), nor was there any loss of rod or cone photoreceptors through P180 (data not shown). These findings suggest that *Mef2c* is dispensable for cone development in mice, and that previous reports of rapid photoreceptor degeneration may have resulted from inadvertent contamination with either the *rd1* allele itself or another common retinal dystrophy mutation (*Chang et al., 2002*).

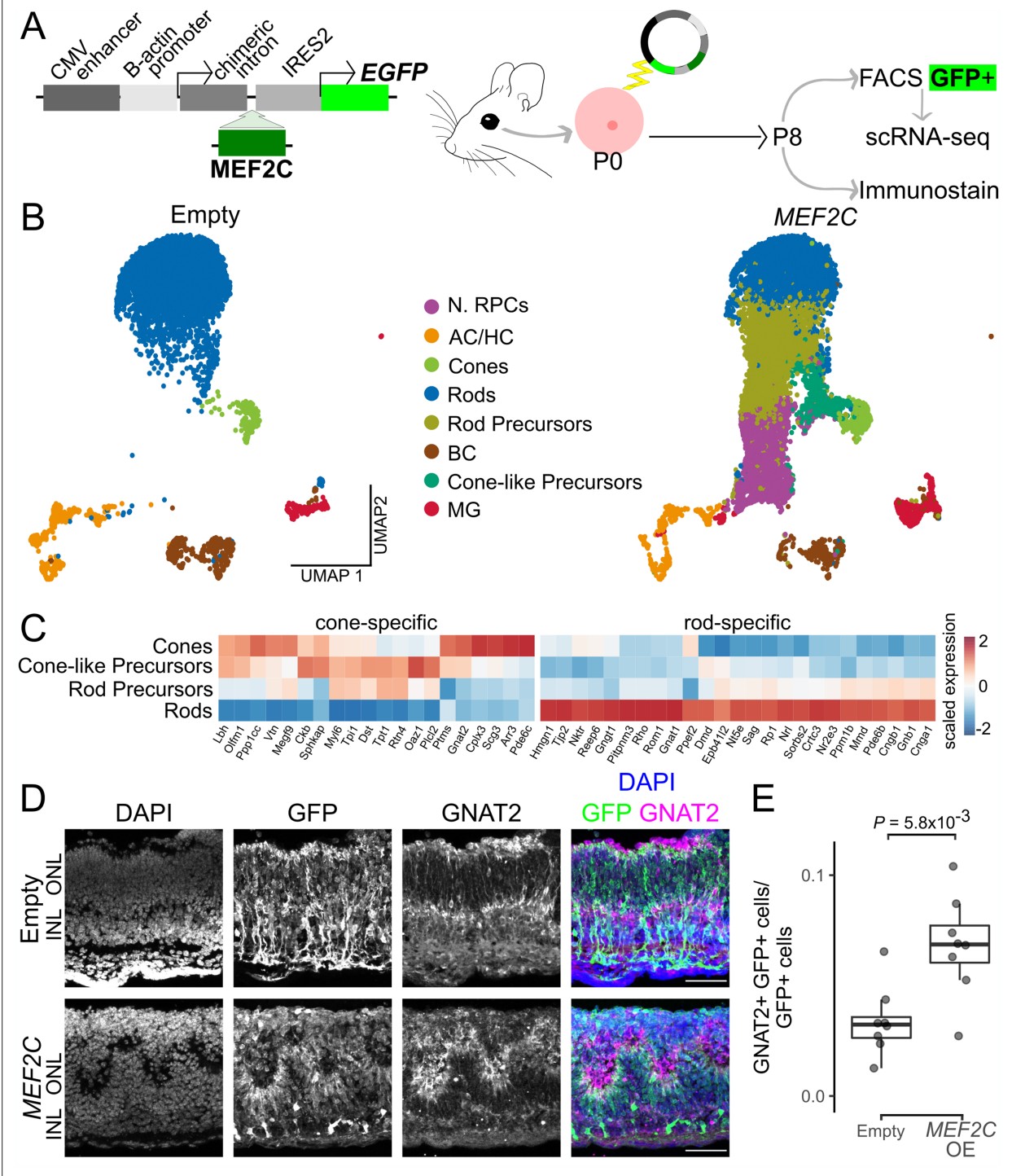

**Figure 4.** Mef2c is sufficient to promote cone-specific gene expression and repress rod-specific gene expression in mouse. (**A**) Diagram of overexpression strategy used to test effects of *MEF2C* overexpression. (**B**) UMAP representation of P8 mouse retinal explants electroporated with a plasmid expressing GFP alone (Empty) or GFP in a bicistronic transcript with human MEF2C (*MEF2C*) (*n* = 7445 cell Empty, 11,949 cells *MEF2C*). Each point represents a single cell and is colored by cell type as determined by clustering and marker gene expression. (**C**) Heatmap of expression for select genes for cones, cone-like photoreceptor precursors, rod photoreceptor precursors, and rods in cells overexpressing *MEF2C*, scaled by gene. (**D**) Immunohistochemistry showing GFP and GNAT2 expression in P8 mouse retinas from Empty and *MEF2C* conditions. Scale bars, 50 µm. (**E**) Box plot of the number of Gnat2+, GFP+ cells divided by the total number of GFP+ cells (*n* = 8 for both conditions). p-values calculated by Wilcoxon rank-sum test. P0, postnatal day 0; P8, postnatal day 8; FACS, fluorescence-activated cell sorting; scRNA-Seq, single-cell RNA sequencing; ONL, outer nuclear layer; INL, inner nuclear layer; DAPI, 4',6-diamidino-2-phenylindole; GFP, green fluorescent protein; OE, overexpression.

*Figure 4 continued on next page*

Figure 4 continued

The online version of this article includes the following figure supplement(s) for figure 4:

**Figure supplement 1.** *Mef2c* overexpression does not obviously impact non-cone cell type proportions.

**Figure supplement 2.** *Mef2c* loss of function does not affect cone cell production or survival.

## Species-specific *cis*-regulatory elements activate expression of cone-promoting genes in late-stage RPCs of 13LGS

These findings demonstrate that, relative to mice, late-stage RPCs and photoreceptor precursors of 13LGS exhibit significantly elevated expression of multiple TFs that promote cone differentiation while inhibiting rod development. To investigate the mechanism underlying this species-specific expansion of expression, we performed bulk CUT&RUN analysis in P5 13LGS retinas and P2 mouse retinas, analyzing stage-matched samples highly enriched for late-stage RPCs and photoreceptor precursors. We profiled histone modifications associated with active promoters and both active and poised enhancers, as well as the photoreceptor-promoting TFs Otx2 and Neurod1. This data was integrated with scATAC-Seq-derived chromatin accessibility regions from late-stage RPCs and photoreceptor precursors (*Figure 5A*, *Supplementary file 9*).

We examined the regulatory elements of genes that exhibited specifically elevated expression in late-stage progenitors and photoreceptor precursors of 13LGS, which were identified in clusters C2 and C3 of the analysis described in *Figure 2D*. Integration of scATAC-Seq and CUT&RUN data showed that 13LGS consistently harbors both greater numbers of accessible chromatin regions associated with markers of both active (H3K27Ac-positive) enhancers and poised (H3K4me1-positive) enhancers for these genes. However, most of these elements mapped to sequences that were not shared between 13LGS and mouse, with intergenic enhancers exhibiting particularly low levels of conservation (*Figure 5B*). In contrast, H3K4me3-positive promoter elements were relatively well-conserved and were found in similar numbers in both species (*Figure 5B*).

Though conserved and species-specific enhancers showed enrichment for binding sites for the broad photoreceptor precursor TFs Otx2 and Neurod1 profiled by CUT&RUN, TFs known to regulate cone differentiation—including *Mef2c*, *Rxrg*, *Thrb*, and *Zic3*—demonstrated far greater motif enrichment in active regulatory elements in 13LGS than in mice, though few of these elements mapped to sequences that were shared between 13LGS and mouse (*Figure 5C, D*, *Supplementary file 10*).

Enhancer elements associated with cone-specific genes in 13LGS are predicted to be directly targeted by a considerably greater number of TFs in late-stage neurogenic RPCs than in mice, as might be expected, given the higher expression levels of these genes. This is particularly evident in enhancers associated with *Thrb* in both species, which are shown to be directly targeted by Otx2 (*Figure 5E, F*). While the *Thrb* promoter is both accessible and bound by Otx2 in mouse photoreceptor precursors, it is predicted to be targeted by significantly fewer TFs compared to the active elements in 13LGS (*Figure 5E, F*). In 13LGS, we identified a pair of species-specific enhancers that are active in late-stage neurogenic RPCs, photoreceptor precursors, and mature cones. These enhancers are predicted to be directly targeted by a combination of factors, including those broadly expressed in developing rods and cones (Otx2, Crx, Neurod1, and Prdm1); cone-promoting factors that are restricted to early-stage RPCs in mice (Zic3, Onecut2, and Pou2f1); and by TFs that selectively promote cone precursor differentiation (Rxrg and Mef2c). The GRNs driving cone specification in 13LGS become progressively more stable over time, as additional TFs are recruited and multiple layers of positive regulation are incorporated (*Figure 5—figure supplement 1A-D*; *Figure 5—figure supplement 2A-C*; *Figure 5—figure supplement 3A-C*).

We conclude that the development of the cone-dominant retina in 13LGS is driven by novel *cis*-regulatory elements that promote heterochronic expression of cone-promoting and rod-inhibiting TFs that are restricted to either early-stage RPCs or postmitotic cone photoreceptor precursors in mice. These *cis*-regulatory elements are activated by a combination of these factors themselves, in conjunction with TFs such as Otx2 and Neurod1, that are expressed in both developing rods and cones. The sustained expression of these factors is maintained by an intricate and progressively expanding network of cross-activating regulatory interactions (*Figure 5G*).

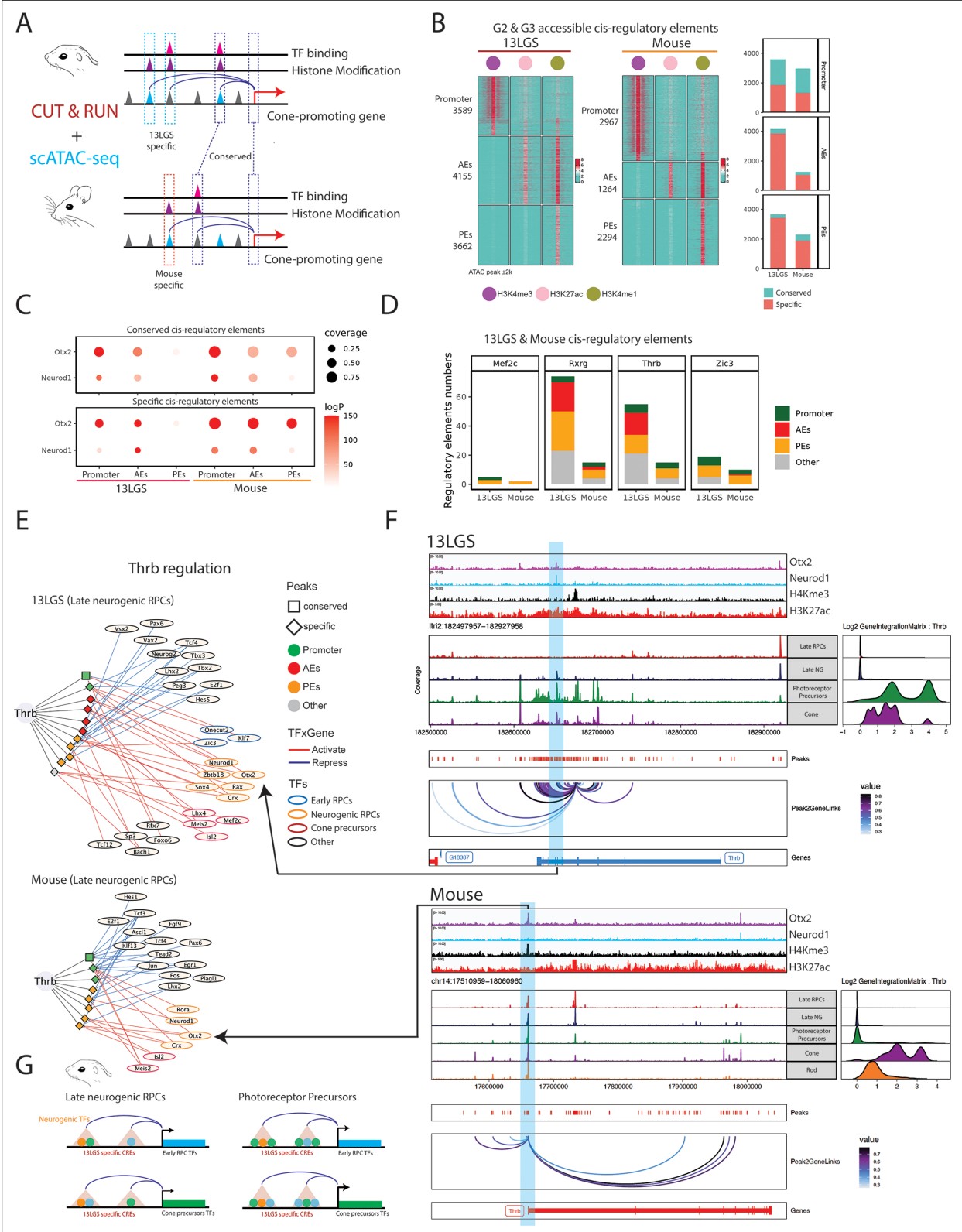

**Figure 5.** Conserved transcription factors (TFs) bind to species-specific enhancers to promote cone specification in 13LGS. (**A**) Schematic illustrating annotation of *cis*-regulatory elements in RPCs and photoreceptor precursors by integration of scATAC-Seq and CUT&RUN 13LGS and mouse datasets. (**B**) Heatmaps show annotated accessible regulatory elements in both 13LGS and mouse. Promoters, activated enhancers (AEs), and poised enhancers (PEs), which are associated with histone markers associated with genes in clusters C2 and C3, which are selectively active in 13LGS RPCs and/or

*Figure 5 continued on next page*

*Figure 5 continued*

photoreceptor precursors. Shading indicates CUT&TAG signal for the corresponding histone modification within 2 kb of the scATAC-Seq peak center. Bar plots displaying the number of each category of regulatory element in each species that are conserved or species-specific. (**C**) Dot plots showing the enrichment of binding sites for Otx2 and Neurod1, TFs which are broadly expressed in both neurogenic RPC and photoreceptor precursors, which are enriched in both conserved *cis*-regulatory elements in both species. (**D**) Bar plots showing the number of conserved and species-specific enhancers per transcription start site (TSS) in four cone-promoting genes between 13LGS and mouse. (**E**) The gene regulatory networks (GRNs) regulating *Thrb* expression in 13LGS and mouse late N. RPCs. (**F**) An example of a Thrb-related regulon and its corresponding scATAC-Seq and CUT&RUN tracks. The arrow indicates the consistent regulatory relationships between GRN prediction and experimental validations. (**G**) The epigenetic model of cone specification in 13LGS and mouse.

The online version of this article includes the following figure supplement(s) for figure 5:

**Figure supplement 1.** Transcriptional regulatory networks controlling *Thrb* expression are more complex in 13LGS than mouse at all developmental stages.

**Figure supplement 2.** Zic3 is regulated by more extensive gene regulatory networks in 13LGS than mouse at all developmental stages.

**Figure supplement 3.** Transcriptional regulatory networks controlling cone-specific gene expression in 13LGS increase in connectivity and complexity as differentiation proceeds.

## Discussion

This study represents the first comprehensive molecular analysis of GRNs that control the development of a cone-dominant retina derived from a rod-dominant ancestral state (*Gerkema et al., 2013*; *Walls, 1942*). Using integrated scRNA- and scATAC-Seq analyses, we systematically examined developmental changes in gene expression and regulation in the cone-dominant 13-lined ground squirrel (13LGS) retina and directly compared these findings to stage-matched data from the rod-dominant mouse. Though BrdU/EdU labeling would be required to unambiguously demonstrate species-specific differences in birthdating, our findings strongly indicate that 13LGS exhibit a selective expansion of the temporal window of cone generation, extending into late stages of neurogenesis. This is mediated by the heterochronic expression of a subset of TFs in late-stage progenitors that both promote cone and repress rod specification. In addition to previously characterized cone-promoting and rod-inhibiting factors such as *Onecut2* and *Pou2f1*, we identified *Zic3*, which is selectively expressed in early-stage RPCs in mice, as also contributing to cone specification. Late-stage 13LGS RPCs also exhibit precocious expression of TFs such as *Thrb*, *Rxrg*, and *Sall3*, which in mice function postmitotically to promote cone differentiation, and in many cases also act in parallel to inhibit rod differentiation (*Swaroop et al., 2010*; *de Melo et al., 2011*).

Transcription of these genes in late-stage RPCs is controlled by 13LGS-specific enhancer elements, which are directly targeted by TFs that are broadly expressed in both early- and late-stage RPCs, such as Otx2, as well as by heterochronically and precociously expressed cone-promoting TFs through a dynamic process of positive feedback and cross-activation. The GRN governing cone specification becomes progressively more complex, incorporating additional TFs over time. These factors include genes expressed in both rod and cone precursors, such as Neurod1 and Prdm1; genes known to promote cone precursor differentiation, such as Sall3; and previously uncharacterized regulators such as Mef2c, which we show can both activate cone-specific genes and repress rod-specific genes.

The evolution of novel morphological features, such as bat wings or the elongated legs of jerboas (*Saxena et al., 2022*; *Cretekos et al., 2008*), is often driven by the emergence of lineage-specific *cis*-regulatory elements, which confer novel spatiotemporal patterns of expression to structurally conserved TFs or secreted patterning regulators. Based on our bioinformatic analysis, the cone-dominant 13LGS retina follows this paradigm, in which species-specific enhancer elements, identified by the overlap of species-specific H3K27Ac peaks and cell type-specific chromatin accessibility, are used to drive both the ectopic heterochronic expression of TFs that regulate neural progenitor temporal identity and precocious expression in retinal progenitors of TFs that normally act to promote cone differentiation in postmitotic precursor cells. These, in turn, promote cone and repress rod photoreceptor specification. The highly interconnected nature of the GRN controlling 13LGS cone specification should make it robust, limiting the impact of the loss of any single enhancer element on cone production. Similarly, the GRN's highly derived nature, with TF-binding combinations not seen in rod-dominant retinas, means that enhancers that are functional in the 13LGS cannot necessarily be expected to behave similarly in mice. In all vertebrates examined to date, including 13LGS,

separate and mutually cross-repressive GRNs specify early- and late-stage temporal identity in retinal progenitors (*Clark et al., 2019*; *Cepko, 2014*; *Lyu et al., 2021*). In mice, cone generation occurs only in progenitors in which the early-stage network is active (*Emerson et al., 2013*; *Javed et al., 2020*; *Lonfat et al., 2021*), whereas in 13LGS, multiple TFs from the early-stage network persist in late-stage RPCs.

This raises the question of whether similar heterochronic shifts in key temporal identity regulators may underlie other cases of rapid evolutionary changes in the rod–cone ratio. Examples in mammals include the development of the cone-dominant retina in primate-like tree shrews and the formation of the rod-free foveola in the primate fovea (*Müller and Peichl, 1989*; *Yuodelis and Hendrickson, 1986*). However, rapid changes in the rod-cone ratio are observed across vertebrate classes (*Peichl, 2005*; *Lamb, 2013*). Significant differences in the relative abundance of other retinal cell types are also observed among species, often correlated with temporal niche adaptation (*Hahn et al., 2023*). Moreover, shifts in the relative ratios of neuronal subtypes in the central nervous system can underlie species-specific behavioral, cognitive, and sensory processes between closely related vertebrate species (*Fernald, 2012*; *Yang et al., 1996*; *Jacobs and Spencer, 1994*; *Young and Wang, 2004*; *Wada et al., 2006*; *Chen et al., 2025*). These variations may also stem from species-specific heterochronic expression of TFs that control temporal identity in the developing brain.

By performing an unbiased and comprehensive analysis of GRNs that are selectively active in late-stage RPCs of 13LGS, we have identified numerous previously uncharacterized factors that are strong candidates for promoting cone specification. In line with previous studies of *Onecut1* and *Pou2f1* (*Emerson et al., 2013*; *Javed et al., 2020*), we find that these factors each activate expression of a subset of cone-specific genes while broadly and potently repress expression of rod-specific genes. Working in mice, we demonstrate that *Zic3* and *Mef2c* behave similarly, activating expression of partially overlapping sets of cone-specific genes and strongly repressing rod differentiation. Indeed, overexpression of *Mef2c* increased the number of both neurogenic RPCs and immature photoreceptor precursors, suggesting that rod differentiation was delayed. Many additional candidates, such as *Hivep2* or *Isl2*, remain to be tested and are likely to serve similar functions. Our analysis suggests that GRNs controlling cone specification are highly redundant, with TFs acting in complex, redundant, and potentially synergistic combinations. This is further supported by our findings on the synergistic effects of combined overexpression of *ZIC3* and *POU21* increasing both the number of DEGs and their level of change in expression relative to the modest changes seen with overexpression of either gene alone (*Figure 3*) and the relatively mild or undetectable phenotypes observed following loss of function of *Zic3* and *Mef2c* (*Figure 3*, *Figure 4—figure supplement 2*), as well as other cone-promoting factors such as *Onecut1* and *Pou2f1* (*Emerson et al., 2013*; *Javed et al., 2020*).

These findings have potentially significant clinical implications, as the loss of cone photoreceptors is the primary trigger for irreversible blindness in hereditary diseases such as retinitis pigmentosa and age-related macular degeneration. Effective strategies for driving cone specification and differentiation are essential for developing cell-based therapies to replace photoreceptors lost to degenerative disease, and the findings of comparative studies like ours can help identify these mechanisms.

## Methods

### Mouse retinal development scRNA- and scATAC-Seq analysis

Raw data and metadata for the mouse scRNA- and scATAC-Seq datasets (*Lyu et al., 2021*; *Clark et al., 2019*) were downloaded from the Gene Expression Omnibus (accession numbers GSE118614 and GSE181251) and analyzed following the Seurat SCT and Signac workflows (*Hao et al., 2021*; *Hafemeister and Satija, 2019*; *Stuart et al., 2021*). Cell type classifications and ages were taken from the downloaded metadata.

### Sample collection

The use of animals for these studies was conducted using protocols approved by the Johns Hopkins Animal Care and Use Committee, the UW Oshkosh Animal Care and Use Committee, and the Northwestern University Animal Care and Use Committee, in compliance with ARRIVE guidelines and the ARVO Statement for the Use of Animals in Ophthalmic and Vision Research, and were performed in accordance with relevant guidelines and regulations.

Timed pregnant CD1 mice were ordered from Charles River Laboratories for the postnatal retinal explant plasmid overexpression analysis. A pair of *Zic3*^+/lox^ (*Jiang et al., 2013*) (JAX stock #023162) female mice and a pair of *Mef2c*^+/lox^ mice (*Vong et al., 2005*) (JAX stock #025556) were ordered from the Jackson Laboratory and outbred to *Chx10Cre-GFP* CD1 mice (*Rowan and Cepko, 2004*). The progeny were bred to produce the *Zic3*^lox/lox^ (or *Zic3*^lox/Y^); *Chx10Cre-GFP*^+/−^, *Zic3*^+/+^ (or *Zic3*^+/Y^); *Chx10Cre-GFP*^+/wt^, *Mef2c*^lox/lox^; *Chx10Cre-GFP*^+wt^, and *Mef2c*^lox/lox^ used in this study. Pups were grown until postnatal day 14 before retinal samples were taken. Mice were euthanized by cervical dislocation.

Embryonic 13-lined ground squirrel (13LGS) retinal samples were collected from timed pregnant captive dams, and embryonic day was confirmed by comparison with mouse Theiler stages. Postnatal 13LGS retinal samples were collected from captive-bred litters with day of birth set as Postnatal Day 0. 13LGS were euthanized by decapitation.

## Sample preservation

Mouse retinas and retinal explants used for immunostaining were fixed in freshly prepared 4% paraformaldehyde in PBS at 4°C for 1 hr (ZIC3) or 2 hr (MEF2C), then briefly washed with PBS for 5 min. After the wash, the explants or retinas were transferred to 30% sucrose in PBS at 4°C overnight. The dehydrated samples were then embedded in VWR Clear Frozen Section Compound, frozen on dry ice, then stored at –80°C.

For single-cell analysis, 13LGS retinas were dissected and either methanol-fixed or flash-frozen as previously shown (*Weir et al., 2021*). Samples were shipped to Johns Hopkins on dry ice and stored at –80°C.

For IHC and RNA-HCR experiments, postnatal 13LGS whole eye cups were fixed in 4% paraformaldehyde (PFA; 157-8, Electron Microscopy Sciences, USA) made in 1X phosphate buffered saline (PBS) after the removal of lens, cornea, and vitreous humor and processed as previously (*Kandoi et al., 2022*; *Yu et al., 2024*). PFA-fixed eye cups were washed three times with 1X PBS at 15 min intervals each, followed by rehydration with 15% and 30% sucrose (S1888-5KG, Sigma Life Science, USA) solutions (made in 1X PBS) until the eyecups had sunk to the bottom of the tube. The whole cups were then equilibrated, embedded, and frozen on dry ice in a sucrose:OCT (4583, Sakura Finetek, Torrance, CA) mixture (2:1) and stored at –80°C.

For CUT&RUN experiments, retinas were dissected out from postnatal 13LGS after the removal of lens, cornea, and vitreous humor. Dissected retinas were immediately flash frozen using liquid nitrogen and shipped to Johns Hopkins University using dry ice and stored at –80°C.

## Retinal explant culture

Retinas were dissected from CD1 mice at postnatal day 0 in warm PBS, then transferred to a microelectroporation chamber. The retinas were electroporated with plasmid solution at 30 V of 50-ms duration with 950-ms intervals (*Matsuda and Cepko, 2008*) using either a pCAGIG plasmid that expresses IRES-GFP (*Matsuda and Cepko, 2004*; *Onishi et al., 2009*) or a test plasmid that expresses the gene of interest as a bicistronic transcript together with IRES-GFP (*Matsuda and Cepko, 2004*). Full-length coding sequences of genes of interest were derivatized into pCAGIG-GW-IRES-GFP (*Venkataraman et al., 2018*), which includes a Gateway Destination Cassette immediately 5′ of IRES sequence. The coding sequences for the Gateway Entry Clones that were derivatized into overexpressed in GW-IRES-GFP constructs correspond to the following accession numbers: human *ZIC3*, NM_003413; human *MEF2C*, XM_011509654; mouse *Onecut1*, NM_008262; human *POU2F1*, NM_002397.

Following harvesting, retinas were flattened by a series of radial cuts and mounted on 0.2 µm Nucleopore Track-Etch membranes on culture media (2 ml DMEM F12, 10% FBS with 0.1% streptomycin/puromycin) in 12-well plates (*Tabata et al., 2004*). In experiments that included a condition where two genes of interest were overexpressed in the same retinal explants, the concentration of plasmids was kept consistent across conditions by supplementing with empty plasmids. For instance, empty condition: 100% control plasmid, condition A: 50% gene A plasmid and 50% control plasmid, condition B: 50% gene A plasmid and 50% control plasmid, and combined condition: 50% gene A plasmid and 50% gene B plasmid.

Explants were incubated at 37°C, 5% $CO_2$ with the media changed every 72 hr. Two-thirds of the old media (1/2 old media in *MEF2C* overexpression) in each well was replaced by slowly adding new

media to the well's wall, avoiding disturbing the membranes. The explants are harvested at P5 (*ZIC3*) or P8 (*MEF2C*) for single-cell RNA sequencing and P8 (*MEF2C*) for immunostaining.

## Retinal sectioning

Mouse embedded samples were sectioned with Leica 3050S cryostat at a thickness of 16 μm and collected on clean Superfrost Plus slides (48311-703, VWR, Radnor, PA). Sections were then stored at –20°C. For postnatal 13LGS, 8–10 μm thick sections were collected and stored at –80°C until use for histological assessments.

## Fluorescence-activated cell sorting and multiplexing

Dissociated single cells from electroporated retinal explants were sorted based on GFP expression on a Sony SH800S cell sorter. GFP+ cells from each condition were labeled and pooled into a single-cell suspension according to the 10x Cell Multiplexing Oligo Labeling for Single Cell RNA Sequencing Protocols with Feature Barcode technology protocol (Rev B) prior to GEM generation and library prep.

## Single-cell library preparation

Retina and explants were dissociated, methanol-fixed cells rehydrated, and nuclei isolated from flash-frozen samples as previously described (*Weir et al., 2021*). GEM generation and library preparation for scRNA-Seq were completed according to the 10x3′ v3.1 user guide (Rev D) using the Chromium Next GEM Single Cell 3′ Kit v3.1 and Dual Index Kit TT Set A and for Cellplex samples according to the 10x Next GEM Single Cell 3′ Reagent Kits v3.1 (Dual Index) with Feature Barcode technology for Cell Multiplexing (Rev A) using the 3′ CellPlex Kit Set A and Dual Index Kit NN Set A. Transposition, GEM generation, and library preparation for scATAC-Seq were completed according to the 10x ATAC v1.1 user guide (Rev D) using the Chromium Next GEM Single Cell ATAC Reagent Kits v1.1 and Single Index Kit N, Set A. Quality control and sequencing were performed by the Johns Hopkins Transcriptomics and Deep Sequencing Core.

## ScRNA- and scATAC-Seq analysis

Sequencing results for 13LGS and mouse scRNA-Seq were processed through the 10x CellRanger pipeline (version 3.1.0), using the 13LGS genome GCF_016881025.1_HiC_Itri_2 (*Fu et al., 2020*), and the output loaded into Seurat (version 4.0.4). Cells from Cellplex-labeled libraries were deconvoluted using the CellRanger multi pipeline (version 6.0). ScRNA-Seq Seurat objects were analyzed following the Seurat SCT workflow (*Hao et al., 2021*; *Hafemeister and Satija, 2019*). Cell type was determined cluster by cluster based on marker gene expression derived from the Seurat function 'FindAllMarkers' using default parameters. Differential gene expression between Cone-like Precursors and Rods was identified using the Seurat function 'FindMarkers'. ScRNA-Seq samples were aggregated using the CellRanger aggr pipeline (version 3.1.0).

Sequencing results for 13LGS scATAC-Seq data were processed through the 10x CellRanger ATAC pipeline (version 1.2.0) and the output loaded into Signac (version 1.4.0) and analyzed according to the Signac workflow. Cell type was determined cluster by cluster based on marker gene expression derived from the Seurat function 'FindAllMarkers' using default parameters and based on transfer learning from the scRNA-Seq samples following the Signac workflow using default parameters. ScATAC-Seq samples were integrated as described (*Lyu et al., 2021*).

## Immunostaining and visualization

Mouse retinal sections were stained with rabbit anti-Arr3 polyclonal IgG (Millipore, No. AB15282, 1:200) or rabbit anti-Gnat2 polyclonal IgG (Invitrogen, PA5-119558, 1:200); chicken anti-GFP polyclonal IgY (Invitrogen, A10262, 1:400); and mouse anti-Nr2e3 monoclonal IgG (R&D Systems, PP-H7223-00, 1:200), goat anti-Nrl polyclonal IgG (R&D Systems, AF2945, 1:200), goat anti-Otx2 polyclonal IgG (R&D Systems, AF1979, 1:400), rabbit anti-Sox9 polyclonal (Millipore, AB5535, 1:400), mouse anti-Tfap2a monoclonal (DSHB, Cat# 3b5, RRID:AB_528084, 1:400), or mouse anti-Lhx1 (DHSB, Cat# 4F2, concentrated) primary antibodies. For the secondary staining, CF633 donkey anti-rabbit IgG polyclonal (Sigma, 1:500); Alexa Fluor 488 goat anti-chicken IgY polyclonal (Invitrogen, 1:500), and Alexa Fluor TM 568 donkey anti-mouse IgG polyclonal (Invitrogen, 1:500) secondary antibodies were used. DAPI was used to counterstain nuclei (1:5000). Stained sections were imaged with a Zeiss

AxioObserver with LSM700 confocal module at 20X resolution within a fixed scanning area of 319.87 $nm^2$. Z-stack images were taken for the stained sections, and cell counting was performed blinded on Z-projections.

For 13LGS IHC experiments, retinal sections were thawed, rehydrated three times with 1X PBS at 15-min intervals, and stained via IHC for identification of retinal cell types. For IHC, sections were permeabilized for 15 min at room temperature with 0.1% Triton X-100 (0694-1L, VWR Life Science, USA) made in 10% normal donkey serum in 1X PBS (NDS, S30-100ML, EMD Millipore Corp, USA). The sections were then blocked for 1 hr at room temperature using 10% normal donkey serum and incubated at 4°C overnight with primary antibodies diluted in 10% normal donkey serum. Primary antibodies utilized for staining 13LGS sections included goat anti-Otx2 polyclonal IgG (R&D Systems, BAF1979, 1:400) and rabbit anti-Mef2c polyclonal (Atlas Antibodies, HPA005533, 1:50). Next day, primary antibodies were removed, sections were washed three times in 1X PBS at room temperature with 15 min interval each and fluorescent conjugated secondary antibodies (donkey anti-mouse IgG Alexa Fluor Plus 647, Thermo Fisher Scientific, A-11055, 1:250; donkey anti-rabbit IgG Alexa Fluor 488, Thermo Fisher Scientific, A-21206, 1:250, and donkey anti-goat IgG Alexa Fluor 555, Thermo Fisher Scientific, A-21432, 1:250) diluted in 10% normal donkey serum were applied to sections and incubated in the dark for 1 hr. Cell nuclei were counterstained with DAPI (1 µg/ml, Roche, 10236276001) for 10 min. The slides were then washed three times in 1X PBS at 5 min intervals and coverslipped using Fluoromount G (17984-25, Electron Microscopy Sciences, USA). Images were acquired using ZEN software on LSM700 confocal microscope (Zeiss, Inc) at 63X oil resolution and processed using ImageJ (*Schindelin et al., 2012*).

## RNA-HCR

The samples were prepared by thawing and dehydrating the 13LGS retinal sections by immersion into a series of ethanol (EtOH) solution—50% EtOH, 70% EtOH, 100% EtOH for 5 min each. The slides were immersed into another fresh 100% EtOH for 5 min and allowed to air dry for 5 min at room temperature. 10 µg/µl of Proteinase K solution (3001-2-60, Zymo Research) was applied on the sections and the slides incubated in a humidified chamber for 10 min at 37°C. The slides were washed by immersion in fresh 1X PBS twice. The slides were dried by blotting the edges on a Kimwipe. The sections were pre-hybridized in a humidified chamber pre-warmed to 37°C for 10 min after adding 200 µl of probe hybridization buffer (240904, HCR Buffer sets, Molecular Instruments). Buffer was removed, and 50–100 µl of DNA probe solutions (1.6 pmol of each probe diluted in probe hybridization buffer) were added to the sections. DNA probes were custom designed using in situ probe generator v0.3.2 at the visual studio code platform for the 13LGS gene targets as shown in *Figure 3* (60 oligos/target at 50 pmol per oligos), obtained from IDT, and reconstituted as 1 µM stock in nuclease-free water. Sections with probe solution were covered with coverslip and incubated overnight (~18–24 hr) at 37°C in a humidified chamber. The next day, the slides were immersed in the probe wash buffer at 37°C to float off the coverslip. Excess probes were removed by incubating the slides at 37°C with pre-heated probe wash buffer (240904, HCR Buffer sets, Molecular Instruments) and 5X SSCT solutions (S6639-1L, Sigma-Aldrich): 75% probe wash buffer/25% 5X SSCT for 15 min; 50% probe wash buffer/50% 5X SSCT for 15 min; 25% probe wash buffer/75% 5X SSCT for 15 min; and 100% 5X SSCT for 15 min. The slides were immersed in 5X SSCT for 5 min at room temperature. The slides were dried by blotting the edges on a Kimwipe. The HCR was pre-amplified by adding 200 µl of amplification buffer (240904, HCR Buffer sets, Molecular Instruments) on top of the sections and incubating the slides for 30 min at room temperature. The solution was removed and 50–100 µl of hairpin solution added before incubating overnight (12–18 hr) in a dark humidified chamber at room temperature. Hairpin solution (Fluorophore labeled for 488/546/647) was prepared by heating (95°C for 90 s) and snap-cooling (in a dark drawer at room temperature for 30 min) 2 µl of 3 µM stocks of h1 and h2 hairpins (6 pmol each) to 100 µl of amplification buffer at room temperature. The next day, the slides were immersed in a 5X SSCT buffer at room temperature to float off the coverslip and counterstained with DAPI (1 µg/ml, Roche, 10236276001) for 10 min, in the dark at room temperature. Excessive hairpins were removed by incubating the slides in a 5X SSCT buffer at room temperature thrice: 2X for 30 min and 1X for 5 min. The slides were coverslipped using Fluoromount G (17984-25, Electron Microscopy Sciences, USA). Images were acquired using ZEN software on an LSM700 confocal microscope (Zeiss, Inc) at 63X oil resolution and processed using ImageJ (NIH, USA).

## CUT&RUN library preparation

Nuclei were isolated from snap-frozen P5 13LGS and P2 mouse retinas for use in CUT&RUN analysis. Two biological replicates were assayed for each species, with each 13LGS replicate consisting of a pool of retinas from two individuals, and each mouse replicate consisting of a pool of retinas from 3 individuals. Nuclei were isolated according to the described protocol (*Tangeman et al., 2025*), using a lysis buffer modified to contain 0.01% Tween-20 and 0.01% Nonidet P40, as well as 1X protease inhibitor cocktail (Roche, 11873580001) in all solutions. Samples were lysed for 5 min in the lysis buffer with gentle trituration throughout, as described. CUT&RUN was performed using the CUTANA ChIC/CUT&RUN Kit (Epicypher, 14-1048). For each reaction, either 500,000 13LGS nuclei or 385,000 mouse nuclei were used. The CUTANA protocol v4.0 protocol was followed with minor modifications. In brief, isolated nuclei were resuspended in CUT&RUN wash buffer and added directly to Concanavalin A beads. Then 0.5 µg of the following antibodies were added to each respective reaction: IgG control (EpiCypher, 13-0042), H3K4me1 (EpiCypher, 13-0057), H3K4me3 (EpiCypher, 13-0041), H3K27ac (EpiCypher, 13-0059), H3K27me3 (EpiCypher, 13-0055), NeuroD1 (Cell Signaling, 62953), Otx2 (R&D Systems, BAF1979), or Otx2 (Atlas Antibodies, HPA000633). H3K9me3 (Abcam, ab8898) was used at 0.41 µg of antibody per reaction. The total CUT&RUN DNA yield, up to 5 ng of total purified DNA, was used for library preparation, using reagents from CUTANA CUT&RUN Library Prep Kit (EpiCypher, 14-1001) or NEBNext Ultra II DNA Library Prep Kit for Illumina (NEB, E7645S). The EpiCypher CUTANA Library Prep Kit User Manual v1.3 was followed and indexing performed using NEBNext Multiplex Oligos for Illumina (NEB, E7395S) or Dual Index Kit TT Set A (10X Genomics, PN-1000215). Final libraries were sequenced at the Novogene sequencing core (Sacramento, CA, USA) on the NovaSeq X Plus (Illumina) to generate paired-end 150 base pair reads.

## CUT&RUN analysis

Sequence reads were trimmed of low-quality bases and adapters using cutadapt v3.5 (*Martin, 2011*) and Trim Galore v0.6.4_dev (*Krueger et al., 2023*), using arguments `---paired --length 20 -e 0.1 --illumina -q 20 --stringency` *3*. Trimmed reads were aligned to mouse genome GRCm38 and 13LGS genome GCF_016881025.1_HiC_Itri_2 using Bowtie2 (*Langmead and Salzberg, 2012*) with arguments *-X 1000* `--very-sensitive-local --no-mixed --no-discordant --no-dovetail`. Unmapped reads with a MAPQ score of 30 or below were filtered from the analysis with SAMtools v1.12 (*Danecek et al., 2021*). Aligned reads were deduplicated using UMI-tools (*Smith et al., 2017*) dedup with parameters -chimeric-pairs="discard" `--unpaired-reads="discard" --paired --ignore-umi`. Technical replicates were collapsed into single biological replicates using samtools merge. For visualization, normalization was applied using deepTools v3.5.1 (*Ramírez et al., 2016*) bamCoverage using arguments `--binSize` *1* `--normalizeUsing` *RPGC* `--effectiveGenomeSize` *2311060300* (13LGS) or `--effectiveGenomeSize` *2651675564* (mouse). For each protein target, peaks were called jointly across replicates against IgG control using macs2 (*Zhang et al., 2008*) using arguments *-f BAMPE* `--keep-dup all` *-q 0.05 -g 2311060300* (13LGS) or *-g 2651675564* (mouse). For mouse samples, peaks on mitochondrial reads and blacklisted ENCODE regions (*Amemiya et al., 2019*) were excluded from analysis.

## Integration of 13LGS and mouse late-stage single-cell datasets

To compare cone development between 13LGS and mouse during the late developmental stages, scRNA-Seq datasets spanning different periods (E21–P8 for 13LGS and E18–P5 for mouse) were integrated. First, the raw gene expression matrices from both species were filtered to retain only conserved genes. Next, 13LGS gene identifiers were mapped to their corresponding mouse gene identifiers as shown in *Supplementary file 2*, and when multiple 13LGS genes corresponded to a single mouse gene, their expression values were summed. The resulting 13LGS and mouse cell-by-gene matrices were directly merged, and the combined raw count matrix was normalized using SCTransform (SCT) before being integrated with Harmony in Seurat. Dimensionality reduction and clustering analyses were then performed on the Harmony dimensions to identify combined cell types (RPCs, N. RPCs, BC/Photoreceptor Precursors, Cones, and Rods). Finally, cell annotations from the scRNA-Seq data were transferred to the scATAC-Seq dataset using the addGeneIntegrationMatrix function in ArchR (version) (*Granja et al., 2021*). Based on the scATAC-Seq clusters and transferred annotations, the

final cell types were manually annotated. For scATAC-Seq datasets, peaks were re-called based on the combined cell types using the addReproduciblePeakSet function in ArchR.

## Identification of differential photoreceptor development genes and their regulatory elements between 13LGS and mouse

To identify DEGs involved in photoreceptor development between the 13LGS and mouse datasets, we utilized the integrated cell-by-gene matrix from the previous integration step. Differential comparisons were performed using the 'FindMarkers' function from the Seurat package with the criteria of p-value <1e−6 and average $\log_2$ fold change >0.35. Specifically, the following comparisons were made: 13LGS_RPCs vs. Mouse_RPCs, 13LGS_N. RPCs vs. Mouse_N. RPCs, 13LGS_BC/Photoreceptor Precursors vs. Mouse_BC/Photoreceptor Precursors, 13LGS_Cone vs. Mouse_Cone, and 13LGS_Cone vs. Mouse_Rod. All DEGs from these comparisons were then merged and clustered according to their expression levels across RPCs, N. RPCs, BC/Photoreceptor precursors, cones, and rods in both species using a $k$-means algorithm ($k$ = 8). Genes in clusters 2 and 3 were subsequently identified as 13LGS-specific cone-promoting genes.

To identify regulatory elements associated with DEGs, scRNA- and scATAC-Seq datasets were first integrated separately for 13LGS and mouse. Next, using the integrated multi-omics data, cells from RPC, NG, BC/Photoreceptor Precursor, Cone, and rod populations were analyzed to calculate peak-to-gene (PtoG) correlations via the 'addPeak2GeneLinks' function in ArchR. For each gene, peaks located within 500 kb that showed a PtoG correlation greater than 0.25 and an FDR below 0.01 were defined as its regulatory elements. Subsequently, the number of regulatory elements per cluster was compared between 13LGS and mouse at the level of conserved gene pairs. For each species, the gene-associated regulatory element counts were normalized by the average number of regulatory elements per gene. Finally, pairwise t-tests were performed to assess the differences in the number of *cis*-regulatory elements across clusters.

## Construction of GRNs

Late-stage cell type-specific GRNs were constructed independently for 13LGS and mouse. To enable a direct comparison between the two networks, we first filtered the cells and genes prior to GRN analysis. For the 13LGS dataset, retinal progenitor cells (RPCs), neurogenic retinal progenitor cells (N. RPCs), bipolar cell/photoreceptor precursors (BC/photoreceptor precursors), and cones were selected. In contrast, the mouse dataset includes RPCs, N. RPCs, BC/photoreceptor precursors, cones, and rods. Additionally, only the conserved genes present in both datasets were retained for GRN construction.

### Identifying *cis*-regulatory elements

For each target gene, three categories of peaks were identified as potential regulatory elements based on the following criteria: (1) Transcription start site (TSS) peaks: Peaks located within a 1-kb region surrounding the target gene's TSS. (2) Gene body peaks: Peaks situated within the target gene body. Only peaks with a PtoG correlation greater than 0.25 and statistically significant (FDR <0.05) were retained. (3) Intergenic peaks: Peaks found within 500 kb of the target gene's TSS, but not overlapping with the TSS or gene body regions of other target genes. These peaks were also required to have a PtoG correlation greater than 0.25 and to be statistically significant (FDR <0.05).

### Predicting TF-binding sites

Cell type-specific TF-binding sites were identified based on TF expression, motif matching, and footprint scores. First, TFs with an average expression level below 0.1 in each cell type were excluded. Next, binding sites for the remaining expressed TFs were determined by matching their motifs— sourced from the TRANSFAC2018 database—to the peak sequences using the matchMotifs function from the motifmatchr package (*Schep, 2017*) (p-value threshold = 5e−5). Second, footprint scores were calculated using merged scATAC-Seq signals for each cell type. The Tn5 insertion sites were extracted using the 'getFragmentsFromProject' function from the ArchR package, converted to BED files, and normalized with TOBIAS software (*Bentsen et al., 2020*). Finally, for each motif's binding region, the average normalized Tn5 insertion signals were calculated by its center (S_center) and flanking regions (S_left and S_right). The flanking region is triple the size of the center region. Motifs were retained if they met the following criteria: S_left − S_center > 0.05 and S_right − S_center > 0.05.

## TF–target correlation

TF–target regulatory scores were computed using the GBM-based GRN inference algorithm via the arboreto package's grnboost2 function (doi: 10.1038/nmeth.4463). Pearson correlations for each TF–target pair were also calculated. Regulatory interactions were classified as positive if the importance score >0.1 and Pearson correlation >0.05, and as negative if the importance score >0.1 and Pearson correlation <−0.05.

## Construction of GRNs

Cell type-specific regulons were constructed by integrating data from steps 1 to 3. The TF–peak–target regulons were classified as activating or repressive based on their TF–target correlations. Duplicated TF–peak–target regulons were removed prior to all downstream analyses. Additionally, TF–target regulons were derived from the TF–peak–target regulons and used to identify cone-promoting TFs.

## Identification of cone-promoting key activator TFs in 13LGS

To identify key TFs promoting cone specification in 13LGS, cone-specific genes were first identified by comparing with other late-stage neurons using the FindMarkers function (cutoffs: $log_2FC$ >0.3 and adjusted p-value <0.01). Next, for each TF, the enrichment of its regulatory targets within the cone-specific genes was assessed using a hypergeometric test via R's phyper function, where the 'population' was defined as the total number of target genes in the positive GRNs, the 'sample' as the number of positive targets for that TF, and the 'successes' as the number of positive targets present in the cone-specific gene list. Additionally, the coverage of each TF was calculated as the proportion of its positive targets that are cone-specific. Finally, TFs with a hypergeometric test p-value <1e−6 and coverage >0.2 were identified as key activators. The top 30 TFs were ranked by the number of their positively regulated cone-specific genes.

## Identification of TSS, aEnhancers, and pEnhancers by CUT&RUN

TSS, bTSS (bivalent TSS), aEnhancer (active enhancer), and pEnhancer (poised enhancer) elements were identified based on the H3K4me3, H3K4me1, and H3K27ac CUT&RUN datasets in conjunction with the candidate *cis*-regulatory elements defined in the previous step using scATAC-Seq analysis. To determine the presence of a histone modification within each candidate *cis*-regulatory element, the corresponding histone peaks were overlapped with these elements, and the average signal intensity for each histone mark was calculated within their regions. A *cis*-regulatory element was considered to exhibit a histone modification if it both overlapped with a histone peak and showed an average signal greater than 2.5. Finally, *cis*-regulatory elements exhibiting H3K4me3+were classified as TSS, those with H3K4me3− and H3K27ac+ as aEnhancer, and those with H3K4me1+ as pEnhancer.

## Identification of conserved regulatory elements

Conserved regulatory elements between 13LGS and mouse were identified by comparing ATAC-Seq peaks (all potential *cis*-regulatory elements), as well as putative regulatory elements associated with promoters, active, and poised enhancers that were associated with conserved gene pairs. For each conserved gene pair, candidate regulatory peaks were collected from both species and paired one-to-one, with each pair comprising one peak from 13LGS and one peak from mouse. Subsequently, the sequence of each 13LGS peak and the forward and reverse complement sequences for each mouse peak were extracted using the Biostrings package in R. Similarity scores for each candidate peak pair were computed with the 'pairwiseAlignment' function using local alignment parameters (type = 'local', gapOpening = −2, gapExtension = −1). To determine the cutoff for conserved peak pairs, a normal model was fitted to all similarity scores using the MASS package's fitdistr function, and a threshold corresponding to a p-value <0.05 under the fitted normal distribution was defined. Peak pairs with scores exceeding this threshold were considered to be conserved.

## ChromVAR analysis

Global TF activity at the single-cell level was evaluated using ChromVAR (*Schep et al., 2017*). The analysis began with the raw cell-by-peak matrix, which was processed with the addGCBias function to adjust for GC content biases. Next, motif information from the TransFac2018 database was

incorporated to generate a TF *z*-score matrix via the matchmotifs and computeDeviations functions. Finally, these *z*-scores were visualized as heatmaps to illustrate TF activity patterns across the cell population.

To identify cell type-specific motifs in 13LGS, the *z*-score matrix generated from ChromVAR was converted into a Seurat object. The FindAllMarkers function from the Signac package was then applied with the following parameters: only.pos=TRUE, mean.fxn=rowMeans, and fc.name = "avg_diff". Finally, motifs with an adjusted *P*-value <0.01 and an average difference (avg_diff) >2 were identified as cell type-specific.

To identify differential motifs associated with photoreceptor development between 13LGS and mouse, the ChromVAR-generated *z*-score matrix was utilized. The following pairwise comparisons were performed: 13LGS_RPCs vs. Mouse_RPCs, 13LGS_N. RPCs vs. Mouse_N. RPCs, 13LGS_BC/Photoreceptor Precursors vs. Mouse_BC/Photoreceptor Precursors, 13LGS_Cone vs. Mouse_Cone, and 13LGS_Cone vs. Mouse_Rod. Motifs exhibiting an adjusted p-value <0.01 and an average difference (avg_diff) >2, with their corresponding genes showing consistent changes, were designated as differential motifs. Subsequently, all differential motifs were merged and clustered via hierarchical clustering based on their average ChromVAR *z*-score levels across RPCs, N. RPCs, BC/photoreceptor precursors, cone, and rod cell types in both species.

## Acknowledgements

We thank J Nathans, A Kolodkin, R Johnston Jr, J Tollkuhn, ML Wilson, DW Kim, and W Yap for comments on the manuscript and V Busa for feedback on figure construction. This work was supported by NIH grants R21EY32281 to SB, F31EY031942 to KW, K00EY036684 to JT, R01EY012141 to SHD, grants from Research to Prevent Blindness to SHD a Bright Focus Macular Degeneration Postdoctoral Fellowship (M2023005F) to SK, and a Pediatric Ophthalmology Career-Starter Research Grant from Knights Templar Eye Foundation, Inc to SK, National Eye Institute (U24EY029891) to DKM, and a Foundation Fighting Blindness to DKM.

## Additional information

### Competing interests

Seth Blackshaw: S.B. is a co-founder, shareholder, and Scientific Advisory Board member of CDI Labs LLC, and has received financial support from Genentech. The other authors declare that no competing interests exist.

### Funding

| Funder | Grant reference number | Author |
| --- | --- | --- |
| National Eye Institute | R21EY32281 | Seth Blackshaw |
| National Eye Institute | F31EY031942 | Kurt Weir |
| National Eye Institute | K00EY036684 | Jared A Tangeman |
| National Eye Institute | R01EY012141 | Steven H DeVries |
| BrightFocus Foundation | M2023005F | Sangeetha Kandoi |
| Knights Templar Eye Foundation | | Sangeetha Kandoi |
| National Eye Institute | U24EY029891 | Dana K Merriman |

The funders had no role in study design, data collection, and interpretation, or the decision to submit the work for publication.

### Author contributions

Kurt Weir, Data curation, Formal analysis, Validation, Investigation, Methodology, Writing – original draft, Writing – review and editing; Pin Lyu, Data curation, Software, Formal analysis, Visualization,

Methodology, Writing – original draft, Writing – review and editing; Sangeetha Kandoi, Validation, Investigation, Visualization, Writing – review and editing; Roujin An, Formal analysis, Validation, Investigation; Nicole Pannullo, Isabella Palazzo, Validation, Investigation, Methodology; Jared A Tangeman, Formal analysis, Validation, Investigation, Writing – review and editing; Jun Shi, Validation, Investigation; Steven H DeVries, Dana K Merriman, Resources, Funding acquisition, Validation, Investigation, Methodology, Project administration, Writing – review and editing; Jiang Qian, Data curation, Software, Formal analysis, Supervision, Writing – original draft, Writing – review and editing; Seth Blackshaw, Conceptualization, Data curation, Supervision, Funding acquisition, Investigation, Methodology, Writing – original draft, Project administration, Writing – review and editing

### Author ORCIDs
Kurt Weir ⓘ https://orcid.org/0000-0002-5957-9227
Pin Lyu ⓘ https://orcid.org/0000-0002-9996-826X
Sangeetha Kandoi ⓘ https://orcid.org/0000-0001-8346-8642
Roujin An ⓘ https://orcid.org/0009-0006-6689-9671
Jared A Tangeman ⓘ https://orcid.org/0000-0002-0224-2893
Seth Blackshaw ⓘ https://orcid.org/0000-0002-1338-8476

### Ethics
The use of animals for these studies was conducted using protocols approved by the Johns Hopkins Animal Care and Use Committee, the UW Oshkosh Animal Care and Use Committee, and the Northwestern University Animal Care and Use Committee, in compliance with ARRIVE guidelines and the ARVO Statement for the Use of Animals in Ophthalmic and Vision Research, all work was performed in accordance with relevant guidelines and regulations under the following approved protocol numbers: MO22M22 (Johns Hopkins), IS00009612 (Northwestern), and 000260-R4-11-11-24 (UW-Oshkosh).

Reviewer #2 (Public review): https://doi.org/10.7554/eLife.108485.3.sa1
Reviewer #3 (Public review): https://doi.org/10.7554/eLife.108485.3.sa2
Author response https://doi.org/10.7554/eLife.108485.3.sa3

---

## Additional files

### Supplementary files
Supplementary file 1. List of cell type-specific patterns of differential gene expression, chromatin accessibility, and motif enrichment in accessible chromatin for 13LGS. Counts for each cell type in each scRNA- and scATAC-Seq library are also shown.

Supplementary file 2. Genes differentially expressed in S- and M-cones in 13LGS.

Supplementary file 3. Related to *Figure 2*, gene ortholog pairs for 13LGS and mouse used in this analysis.

Supplementary file 4. Related to *Figure 2*, genes and accessible transcription factor motifs showing differential activity between 13LGS and mouse in late-stage progenitors and differentiating photoreceptors.

Supplementary file 5. Related to *Figure 2*, top cone-promoting transcription factors active in 13LGS.

Supplementary file 6. Related to *Figure 2*, list of genes differentially expressed following overexpression of *ZIC3*, *Onecut1*, *POU2F1*, and *ZIC3+POU2F1* as detected by scRNA-Seq and proportions of major cell types observed in each sample.

Supplementary file 7. Related to *Figure 3*, a list of genes differentially expressed in *Chx10-Cre;Zic3^{lox/lox}* retina and proportions of major cell types observed in each sample.

Supplementary file 8. Related to *Figure 4*, list of genes differentially expressed following overexpression of *MEF2C* and proportions of major cell types observed in each sample.

Supplementary file 9. Related to *Figure 5*, *cis*-regulatory elements inferred by scATAC-Seq and CUT&TAG for genes in clusters 2 and 3 in 13LGS and mouse.

Supplementary file 10. Related to *Figure 5*, active and poised enhancers associated with *Mef2c*, *Rxrg*, *Thrb*, and *Zic3* in 13LGS and mouse.

MDAR checklist

## Data availability

All scRNA-Seq, scATAC-Seq, and CUT&RUN data are available as GEO: GSE303632. The custom scripts used in this paper: https://github.com/lp871/13LGS-Development (copy archived at *lp871, 2025*).

The following dataset was generated:

| Author(s) | Year | Dataset title | Dataset URL | Database and Identifier |
|---|---|---|---|---|
| Weir K, Lyu P, Kandoi S, An R, Pannullo N, Palazzo I, Tangeman JA, Shi J, DeVries SH, Merriman DK, Qian J, Blackshaw S | 2025 | Heterochronic transcription factor expression drives cone-dominant retina development in 13-lined ground squirrels | https://www.ncbi.nlm.nih.gov/geo/query/acc.cgi?acc=GSE295358 | NCBI Gene Expression Omnibus, GSE295358 |

The following previously published datasets were used:

| Author(s) | Year | Dataset title | Dataset URL | Database and Identifier |
|---|---|---|---|---|
| Lyu P, Hoang T, Santiago CP, Thomas ED, Timms AE, Appel H, Gimmen M, Le N, Jiang L, Kim DW, Chen S, Espinoza D, Telger AE, Weir K, Clark BS, Cherry TJ, Qian J, Blackshaw S | 2021 | Gene regulatory networks controlling temporal patterning, neurogenesis and cell fate specification in the mammalian retina | https://www.ncbi.nlm.nih.gov/geo/query/acc.cgi?acc=GSE181251 | NCBI Gene Expression Omnibus, GSE181251 |
| Clark BS, Stein-O'Brien GL, Shiau F, Cannon GH, Davis E, Sherman T, Rajaii F, James-Esposito RE, Gronostajski RM, Fertig EJ, Goff LA, Blackshaw S | 2019 | Single-cell RNA-Seq Analysis of Retinal Development Identifies NFI Factors as Regulating Mitotic Exit and Late-Born Cell Specification | https://www.ncbi.nlm.nih.gov/geo/query/acc.cgi?acc=GSE118614 | NCBI Gene Expression Omnibus, GSE118614 |

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

# Appendix 1

### Appendix 1—key resources table

| Reagent type (species) or resource | Designation | Source or reference | Identifiers | Additional information |
|---|---|---|---|---|
| Antibody | rabbit anti-Arr3 polyclonal | Millipore | No. AB15282 | 1:200 dilution |
| Antibody | rabbit anti-Gnat2 polyclonal | Invitrogen | PA5-119558 | 1:200 dilution |
| Antibody | chicken anti-GFP polyclonal | Invitrogen | A10262 | 1:400 dilution |
| Antibody | mouse anti-Nr2e3 monoclonal | R&D Systems | PP-H7223-00 | 1:200 dilution |
| Antibody | goat anti-Nrl polyclonal | R&D Systems | AF2945 | 1:200 dilution |
| Antibody | goat anti-Otx2 polyclonal | R&D Systems | AF1979 | 1:400 dilution |
| Antibody | rabbit anti-Sox9 polyclonal | Millipore | AB5535 | 1:400 dilution |
| Antibody | mouse anti-Tfap2a monoclonal | DSHB | Cat# 3b5, RRID:AB_528084 | 1:400 dilution |
| Antibody | mouse anti-Lhx1 monoclonal | DHSB | Cat# 4F2/concentrate | 1:400 dilution |
| Antibody | goat anti-Otx2 polyclonal | R&D Systems | BAF1979 | 1:200 dilution |
| Antibody | rabbit anti-Mef2c polyclonal | Atlas Antibodies | HPA005533 | 1:200 dilution |
| Antibody | donkey anti-mouse Alexa Fluor Plus 647 polyclonal | Thermo Fisher Scientific | A-11055 | 1:250 dilution |
| Antibody | donkey anti-rabbit Alexa Fluor 488 polyclonal | Thermo Fisher Scientific | A-21206 | 1:250 dilution |
| Antibody | donkey anti-goat IgG Alexa Fluor 555 polyclonal | Thermo Fisher Scientific | A-21432 | 1:250 dilution |
| Antibody | CF633 donkey anti-rabbit polyclonal | Sigma | SAB4600132-50UL | 1:250 dilution |
| Antibody | goat anti-chicken Alexa Fluor 488 polyclonal | Invitrogen | A-11039 | 1:250 dilution |
| Antibody | donkey anti-mouse Alexa Fluor 568 polyclonal | Invitrogen | A10037 | 1:250 dilution |
| Antibody | CUTANA IgG Negative Control Antibody for CUT&RUN and CUT&Tag | EpiCypher | 13-0042 | 0.5 µg/reaction |
| Antibody | H3K4me1 Antibody, SNAP-Certified for CUT&RUN and CUT&Tag | EpiCypher | 13-0057 | 0.5 µg/reaction |
| Antibody | H3K4me3 Antibody, SNAP-Certified for CUT&RUN | EpiCypher | 13-0041 | 0.41 µg/reaction |
| Antibody | H3K27ac Antibody, SNAP-Certified for CUT&RUN and CUT&Tag | EpiCypher | 13-0059 | 0.5 µg/reaction |
| Antibody | H3K27me3 Antibody, SNAP-Certified for CUT&RUN and CUT&Tag | EpiCypher | 13-0055 | 0.5 µg/reaction |
| Antibody | NeuroD1 (E3E4F) Rabbit mAb | Cell Signaling | 62953 | 0.5 µg/reaction |
| Antibody | rabbit Anti-OTX2 Antibody polyclonal | Atlas Antibodies | HPA000633 | 0.5 µg/reaction |
| Antibody | Anti-Histone H3 (tri methyl K9) antibody – ChIP Grade | Abcam | ab8898 | 0.5 µg/reaction |
| Biological sample | 13LGS retinal samples | This study | N/A | Animal colony in De Vries and Merriman labs |
| Commercial assay/kit | Chromium Next GEM Single Cell 3' Kit v3.1 | 10X Genomics | PN-1000268 | |

*Appendix 1 Continued on next page*

*Appendix 1 Continued*

| Reagent type (species) or resource | Designation | Source or reference | Identifiers | Additional information |
|---|---|---|---|---|
| Commercial assay/kit | Dual Index Kit TT Set A | 10X Genomics | PN-1000215 | |
| Commercial assay/kit | Chromium Next GEM Single Cell ATAC Reagent Kits v1.1 | 10X Genomics | PN-1000175 | |
| Commercial assay/kit | Single Index Kit N, Set A | 10X Genomics | PN-1000212 | |
| Commercial assay/kit | 3' CellPlex Kit Set A | 10X Genomics | PN-1000261 | |
| Commercial assay/kit | Dual Index Kit NN Set A | 10X Genomics | PN-1000243 | |
| Commercial assay/kit | CUTANA ChIC/CUT&RUN Kit | Epicypher | 14-1048 | |
| Commercial assay/kit | CUTANA CUT&RUN Library Prep Kit | EpiCypher | 14-1001 | |
| Commercial assay/kit | NEBNext Ultra II DNA Library Prep Kit for Illumina | NEB | E7645S | |
| Commercial assay/kit | NEBNext Multiplex Oligos for Illumina | NEB | E7395S | |
| Other | Mouse retinal development scRNA-seq data | *Clark et al., 2019* | GEO: GSE118614 | |
| Other | Mouse retinal development scATAC-seq data | *Lyu et al., 2021* | GEO: GSE181251 | |
| Other | 13LGS retinal development scRNA-seq, scATAC-seq data 13LGS and Mouse Cut&Run data Mouse *Zic3*, *Pou2f1*, *Onecut1*, *Mef2c* overexpression and *Zic3* cKO scRNA-seq data | This study | GEO: GSE295358 | |
| Other | 13LGS genome GCF_016881025.1_HiC_Itri_2 | *Rhie et al., 2021* | https://ftp.ncbi.nlm.nih.gov/genomes/all/GCF/016/881/025/GCF_016881025.1_HiC_Itri_2/ | |
| Genetic reagent | *Mus musculus: Zic3*+/lox: B6;129-Zic3tm2.1Jwb/J | The Jackson Lab | RRID:IMSR_JAX:023162 | *Jiang et al., 2013* |
| Genetic reagent | *Mus musculus: Mef2c*+/lox: Mef2ctm1Jjs/J | The Jackson Lab | RRID:IMSR_JAX:025556 | *Vong et al., 2005* |
| Genetic reagent | *Mus musculus: Chx10Cre-GFP* | The Jackson Lab | RRID:IMSR_MGI:3051137 | *Rowan and Cepko, 2004* |
| Recombinant DNA reagent | *ZIC3* Gateway Ultimate Human ORF Clone | Thermo Fisher Scientific | NM_003413 (coding sequence only) | |
| Recombinant DNA reagent | *POU2F1* Gateway Ultimate Human ORF Clone | Thermo Fisher Scientific | XM_011509654 (coding sequence only) | |
| Recombinant DNA reagent | *Onecut1* Mouse ORF Clone | This study | NM_008262 (coding sequence only) | Available upon request |
| Recombinant DNA reagent | *MEF2C* Gateway Ultimate Human ORF Clone | Thermo Fisher Scientific | NM_002397 (coding sequence only) | |
| Recombinant DNA reagent | Plasmid: pCAGIG IRES-GFP | Addgene plasmid #11159 | N/A | *Matsuda and Cepko, 2004* |
| Recombinant DNA reagent | Plasmid: pCAGIG-GW-IRES-GFP | *Venkataraman et al., 2018* | N/A | Available upon request |
| Software, algorithm | CellRanger Pipeline (version 3.1.0) | 10X Genomics | https://www.10xgenomics.com/support/software | |

*Appendix 1 Continued on next page*

*Appendix 1 Continued*

| Reagent type (species) or resource | Designation | Source or reference | Identifiers | Additional information |
|---|---|---|---|---|
| Software, algorithm | CellRanger Multi Pipeline (version 6.0) | 10X Genomics | https://www.10xgenomics.com/support/software | |
| Software, algorithm | CellRanger Aggr Pipeline (version 3.1.0) | 10X Genomics | https://www.10xgenomics.com/support/software | |
| Software, algorithm | CellRanger ATAC pipeline (version 1.2.0) | 10X Genomics | https://www.10xgenomics.com/support/software | |
| Software, algorithm | Seurat (version 4.0.4) | *Hao et al., 2021* | https://satijalab.org/seurat/articles/install_v5.html | |
| Software, algorithm | Signac (version 1.4.0) | *Stuart et al., 2021* | https://satijalab.org/signac | |
| Software, algorithm | cutadapt (version 3.5) | *Martin, 2011* | https://cutadapt.readthedocs.io/en/stable/ | |
| Software, algorithm | Trim Galore (version 0.6.4_dev) | *Krueger et al., 2023* | https://github.com/FelixKrueger/TrimGalore RRID:SCR_011847 | |
| Software, algorithm | Bowtie2 | *Langmead and Salzberg, 2012*; *Langmead et al., 2026* | https://github.com/BenLangmead/bowtie2; RRID:SCR_016368 | |
| Software, algorithm | SAMtools (version 1.12) | *Danecek et al., 2021* | https://www.htslib.org | |
| Software, algorithm | UMI-tools | *Smith et al., 2017* | https://umi-tools.readthedocs.io/en/latest/index.html | |
| Software, algorithm | deepTools (version 3.5.1) | *Ramírez et al., 2016* | https://deeptools.readthedocs.io/en/latest/ | |
| Software, algorithm | macs2 | *Zhang et al., 2008* | https://pypi.org/project/MACS2/ | |
| Software, algorithm | ArchR | *Granja et al., 2021* | https://www.archrproject.com | |
| Software, algorithm | ChromVAR | *Schep et al., 2017* | https://greenleaflab.github.io/chromVAR/ | |
| Software, algorithm | ImageJ | *Schindelin et al., 2012* | https://imagej.net/ij/ | |

