## [Editor Report · eLife Assessment]

This **important** study investigates why the 13-lined ground squirrel (13LGS) retina is unusually rich in cone photoreceptors, the cells responsible for color and daylight vision. The authors perform deep transcriptomic and epigenetic comparisons between the mouse and the 13-lined ground squirrel (13LGS) to provide **convincing** evidence that identifies mechanisms that drive rod vs cone-rich retina development. Overall, this key question is investigated using an impressive collection of new data, cross-species analysis, and subsequent in vivo experiments.

---

## [Referee Report · Reviewer #2 (Public review)]

Summary:

This paper aims to elucidate the gene regulatory network governing the development of cone photoreceptors, the light-sensing neurons responsible for high acuity and color vision in humans. The authors provide a comprehensive analysis through stage-matched comparisons of gene expression and chromatin accessibility using scRNA-seq and scATAC-seq from the cone-dominant 13-lined ground squirrel (13LGS) retina and the rod-dominant mouse retina. The abundance of cones in the 13LGS retina arises from a dominant trajectory from late retinal progenitor cells (RPCs) to photoreceptor precursors and then to cones, whereas only a small proportion of rods are generated from these precursors.

Strengths:

The paper presents intriguing insights into the gene regulatory network involved in 13LGS cone development. In particular, the authors highlight the expression of cone-promoting transcription factors such as Onecut2, Pou2f1, and Zic3 in late-stage neurogenic progenitors, which may be driven by 13LGS-specific cis-regulatory elements. The authors also characterize candidate cone-promoting genes Zic3 and Mef2C, which have been previously understudied. Overall, I found that the across-species analysis presented by this study is a useful resource for the field.

Comments on Revision:

The authors have addressed my questions, and the revised text now presents their findings more clearly.

---

## [Referee Report · Reviewer #3 (Public review)]

Summary:

The authors perform deep transcriptomic and epigenetic comparisons between mouse and 13-lined ground squirrel (13LGS) to identify mechanisms that drive rod vs cone rich retina development. Through cross species analysis the authors find extended cone generation in 13LGS, gene expression within progenitor/photoreceptor precursor cells consistent with lengthened cone window, and differential regulatory element usage. Two of the transcription factors, Mef2c and Zic3, were subsequently validated using OE and KO mouse lines to verify role of these genes in regulating competence to generate cone photoreceptors.

Strengths:

Overall, this is an impactful manuscript with broad implications toward our understanding of retinal development, cell fate specification, and TF network dynamics across evolution and with the potential to influence our future ability to treat vision loss in human patients. The generation of this rich new dataset profiling the transcriptome and epigenome of the 13LGS is a tremendous addition to the field that assuredly will be useful for numerous other investigations and questions of a variety of interests. In this manuscript, the authors use this dataset and compare to data they previously generated for mouse retinal development to identify 2 new regulators of cone generation and shed insights onto their regulation and their integration into the network of regulatory elements within the 13LGS compared to mouse.

The authors have done considerable work to address reviewer concerns from the first draft. The current version of the manuscript is strong and supports the claims.

---

## [Author Response]

The following is the authors’ response to the original reviews.

**Public Reviews:**

**Reviewer #1 (Public review):**
SummaryIn this manuscript, Weir et al. investigate why the 13-lined ground squirrel (13LGS) retina is unusually rich in cone photoreceptors, the cells responsible for color and daylight vision. Most mammals, including humans, have rod-dominant retinas, making the 13LGS retina both an intriguing evolutionary divergence and a valuable model for uncovering novel mechanisms of cone generation. The developmental programs underlying this adaptation were previously unknown.Using an integrated approach that combines single-cell RNA sequencing (scRNAseq), scATACseq, and histology, the authors generate a comprehensive atlas of retinal neurogenesis in 13LGS. Notably, comparative analyses with mouse datasets reveal that in 13LGS, cones can arise from late-stage neurogenic progenitors, a striking contrast to mouse and primate retinas, where late progenitors typically generate rods and other late-born cell types but not cones. They further identify a shift in the timing (heterochrony) of expression of several transcription factors.Further, the authors show that these factors act through species-specific regulatory elements. And overall, functional experiments support a role for several of these candidates in cone production.StrengthsThis study stands out for its rigorous and multi-layered methodology. The combination of transcriptomic, epigenomic, and histological data yields a detailed and coherent view of cone development in 13LGS. Cross-species comparisons are thoughtfully executed, lending strong evolutionary context to the findings. The conclusions are, in general, well supported by the evidence, and the datasets generated represent a substantial resource for the field. The work will be of high value to both evolutionary neurobiology and regenerative medicine, particularly in the design of strategies to replace lost cone photoreceptors in human disease.Weaknesses(1) Overall, the conclusions are strongly supported by the data, but the paper would benefit from additional clarifications. In particular, some of the conclusions could be toned down slightly to reflect that the observed changes in candidate gene function, such as those for Zic3 by itself, are modest and may represent part of a more complex regulatory network.

We have revised the text to qualify these conclusions as suggested.

“Zic3 promotes cone-specific gene expression and is necessary for generating the full complement of cone photoreceptors”

“Pou2f1 overexpression upregulated an overlapping but distinct, and larger, set of cone-specific genes relative to Zic3, while also downregulating many of the same rod-specific genes, often to a greater extent (Fig. 3C).”

“This resulted in a statistically significant ~20% reduction in the density of cone photoreceptors in the mutant retina (Fig. 3E,F), while the relative numbers of rods and horizontal cells remained unaffected (Fig. S4A-D).”

“Our analysis suggests that gene regulatory networks controlling cone specification are highly redundant, with transcription factors acting in complex, redundant, and potentially synergistic combinations. This is further supported by our findings on the synergistic effects of combined overexpression of Zic3 and Pou2f1 increasing both the number of differentially expressed genes and their level of change in expression relative to the modest changes seen with overexpression of either gene alone (Fig. 3) and the relatively mild or undetectable phenotypes observed following loss of function of Zic3 and Mef2c (Fig. 3, Fig. S6), as well as other cone-promoting factors such as Onecut1 and Pou2f1[18,19].“

(2) Additional explanations about the cell composition of the 13LGS retina are needed. The ratios between cone and rod are clearly detailed, but do those lead to changes in other cell types?

The 13LGS retina, like most cone-dominant retinas, shows relatively lower numbers of rod and cone photoreceptors (~20%) than do nocturnal species such as mice (~80%). The difference is made up by increased numbers of inner retinal neurons and Muller glia. While rigorous histological quantification of the abundance of inner retinal cell types has not yet been performed for 13LGS, we can estimate these values using our snATAC-Seq data. These numbers are provided in Table ST1, and are now discussed in the text.

(3) Could the lack of a clear trajectory for rod differentiation be just an effect of low cell numbers for this population?

This is indeed likely to be the case. This is now stated explicitly in the text.

“However, no clear trajectory for rod differentiation was detected, likely due to the very low number of rod cells detected prior to P17 (Fig. 2A).”

(4) The immunohistochemistry and RNA hybridization experiments shown in Figure S2 would benefit from supporting controls to strengthen their interpretability. While it has to be recognized that performing immunostainings on non-conventional species is not a simple task, negative controls are necessary to establish the baseline background levels, especially in cases where there seems to be labeling around the cells. The text indicates that these experiments are both immunostainings and ISH, but the figure legend only says "immunohistochemistry". Clarifying these points would improve readers' confidence in the data.

The figure legend has been corrected, and negative controls for P24 have been added. The figure legend has been modified as follows:

“Fluorescent in situ hybridization showing co-expression of (A) Pou2f1 and Otx2 or (B) Zic3, Rxrg, and Otx2 in P1, P5, P10, and P24 retinas. Insets show higher power images of highlighted areas. (C) Zic3, Rxrg, and Otx2 fluorescent in situ hybridization from P24 with matched (C’) negative controls. (D) Pou2f1 and Otx2 fluorescent in situ hybridization from P24 with matched (D’) negative controls. (E) Quantification of the fraction of Otx2-positive cells in the outer neuroblastic layer (P1, P5) and ONL (P10, P24) that also express Zic3. (F) Immunohistochemical analysis Mef2c and Otx2 expression in P1, P5, P10, and P24 retinas. (G) Mef2c and Otx2 immunohistochemistry from P24 with matched (G’) negative controls. Negative controls for fluorescent in situ hybridization omit the probe and for immunohistochemistry omit primary antibodies. Scale bars, 10 µm (S2A-F), 50 µm (S2G) and 5 µm (inset). Cell counts in E were analyzed using one-way ANOVA analysis with Sidak multiple comparisons test and 95% confidence interval. ** = p <0.01, **** = p <0.0001, and ns = non-significant. N=3 independent experiments.”

(5) Figure S3: The text claims that overexpression of Zic3 alone is sufficient to induce the conelike photoreceptor precursor cells as well as horizontal cell-like precursors, but this is not clear in Figure S3A nor in any other figure. Similarly, the effects of Pou2f1 overexpression are different in Figure S3A and Figure S3B. In Figure S3B, the effects described (increased presence of cone-like and horizontal-like precursors) are very clear, whereas it is not in Figure S3A. How are these experiments different?

These UMAP data represent two independent experiments. Total numbers and relative fractions of each cell type are now included in Table ST5.

In these experiments, cone-like precursors were identified by both cell type clustering and differential gene expression. Cells from all conditions were found in the cone-like precursor cluster. However, cells electroporated with a plasmid expressing GFP alone only showed GFP as a differentially expressed gene, identifying them most likely as GFP+ rods. In contrast, Zic3 overexpression resulted in increased expression of cone-specific genes and decreased expression of rod-specific genes in both cone-like precursors and rods relative to controls electroporated with GFP alone. Cell type proportions across independent overexpression singlecell experiments could be influenced by a number of factors, including electroporation efficiency and ex vivo growth conditions.

(6) The analyses of Zic3 conditional mutants (Figure S4) reveal an increase in many cone, rod, and pan-photoreceptor genes with only a reduction in some cone genes. Thus, the overall conclusion that Zic3 is essential for cones while repressing rod genes doesn't seem to match this particular dataset.

We observe that loss of function of Zic3 in developing retinal progenitors leads to a reduction in the total number of cones (Fig. 4E,F). In Fig. S4, we investigate how gene expression is altered in both the remaining cones and in other retinal cell types. We only observed significant changes in mutant cones and Muller glia relative to controls. We observe a mixed phenotype in cones, with a subset of cone-specific genes downregulated (notably including Thrb), a subset of others upregulated (including Opn1sw). We also find that genes expressed both in rods and cones, as well as rod-specific genes, are downregulated in cKO cones. Since rods are fragile cells that are located immediately adjacent to cones, some level of contamination of rod-specific genes is inevitable in single-cell analysis of dissociated cones (c.f. PMID: 31128945, 34788628), and this reduced level of rod contamination could result from altered adhesion between mutant rods and cones. In mutant Muller glia, in contrast, we see a broad decrease in expression of Muller glia-specific genes, which likely reflects the indirect effects of Zic3 loss of function in retinal progenitors, and an upregulation of both broadly photoreceptor-specific genes and a subset of rod-specific genes, which may also result from altered adhesion between Muller glia and rods.

This is consistent with the conclusions in the text, although we have both modified the text and included heatmaps showing downregulation of rod-specific genes in mutant cones, to clarify this finding.

“In addition, we observe a broad decrease in expression of genes expressed at high levels in both cones and rods (Rpgrip1, Drd4) and rod-specific genes (Rho, Cnga1, Pde6b) in mutant cones (Fig. S4F). Since rods are fragile cells that are located immediately adjacent to cones, some level of contamination of rod-specific genes is inevitable in single-cell analysis of dissociated cones (c.f. PMID: 31128945, 34788628), and this reduced level of rod contamination could result from altered adhesion between mutant rods and cones. In contrast, increased expression of rod-specific genes (Rho, Nrl, Pde6g, Gngt1) and pan-photoreceptor genes (Crx, Stx3, Rcvrn) was observed in Müller glia (Fig. S4G), which may likewise result from altered adhesion between Muller glia and rods. Finally, several Müller glia-specific genes were downregulated, including Clu, Aqp4, and Notch pathway components such as Hes1 and Id3, with the exception of Hopx, which was upregulated (Fig. S4G). This likely reflects the indirect effects of Zic3 loss of function in retinal progenitors. These findings indicate that Zic3 is essential for the proper expression of photoreceptor genes in cones while also playing a role in regulating expression of Müller glia-specific genes.”

(7) Throughout the text, the authors used the term "evolved". To substantiate this claim, it would be important to include sequence analyses or to rephrase to a more neutral term that does not imply evolutionary inference.

We have modified the text as requested to replace “evolved” and “evolutionarily conserved” where possible, with examples of revised text listed below:

“These results demonstrate that modifications to gene regulatory networks underlie the development of cone-dominant retina,...”

“Our results demonstrate that heterochronic expansion of the expression of transcription factors that promote cone development is a key event in the development of the cone-dominant 13LGS retina.”

“Conserved patterns of motif accessibility, identified using ChromVAR and theTRANSFAC2018 database, (Fig. S1F, Table ST1)...”

“However, most of these elements mapped to sequences that were not shared between 13LGS and mouse, with intergenic enhancers exhibiting particularly low levels of conservation (Fig. 5B).”

“We conclude that the development of the cone-dominant retina in 13LGS is driven by novel cisregulatory elements…”

“Based on our bioinformatic analysis, the cone-dominant 13LGS retina follows this paradigm, in which species-specific enhancer elements…”

“Dot plots showing the enrichment of binding sites for Otx2 and Neurod1, TFs which are broadly expressed in both neurogenic RPC and photoreceptor precursors, which are enriched in both conserved cis-regulatory elements in both species. (D) Bar plots showing the number of conversed and species-specific enhancers per TSS in four cone-promoting genes between 13LGS and mouse.”

**Reviewer #2 (Public review):**
Summary:This paper aims to elucidate the gene regulatory network governing the development of cone photoreceptors, the light-sensing neurons responsible for high acuity and color vision in humans. The authors provide a comprehensive analysis through stage-matched comparisons of gene expression and chromatin accessibility using scRNA-seq and scATAC-seq from the conedominant 13-lined ground squirrel (13LGS) retina and the rod-dominant mouse retina. The abundance of cones in the 13LGS retina arises from a dominant trajectory from late retinal progenitor cells (RPCs) to photoreceptor precursors and then to cones, whereas only a small proportion of rods are generated from these precursors.Strengths:The paper presents intriguing insights into the gene regulatory network involved in 13LGS cone development. In particular, the authors highlight the expression of cone-promoting transcription factors such as Onecut2, Pou2f1, and Zic3 in late-stage neurogenic progenitors, which may be driven by 13LGS-specific cis-regulatory elements. The authors also characterize candidate cone-promoting genes Zic3 and Mef2C, which have been previously understudied. Overall, I found that the across-species analysis presented by this study is a useful resource for the field.Weaknesses:The functional analysis on Zic3 and Mef2C in mice does not convincingly establish that these factors are sufficient or necessary to promote cone photoreceptor specification. Several analyses lack clarity or consistency, and figure labeling and interpretation need improvement.

We have modified the text and figures to more clearly describe the observed roles of Zic3 and Mef2c in cone photoreceptor development as detailed in our responses to reviewer recommendations.

**Reviewer #3 (Public review):**
Summary:The authors perform deep transcriptomic and epigenetic comparisons between mouse and 13lined ground squirrel (13LGS) to identify mechanisms that drive rod vs cone-rich retina development. Through cross-species analysis, the authors find extended cone generation in 13LGS, gene expression within progenitor/photoreceptor precursor cells consistent with a lengthened cone window, and differential regulatory element usage. Two of the transcription factors, Mef2c and Zic3, were subsequently validated using OE and KO mouse lines to verify the role of these genes in regulating competence to generate cone photoreceptors.Strengths:Overall, this is an impactful manuscript with broad implications toward our understanding of retinal development, cell fate specification, and TF network dynamics across evolution and with the potential to influence our future ability to treat vision loss in human patients. The generation of this rich new dataset profiling the transcriptome and epigenome of the 13LGS is a tremendous addition to the field that assuredly will be useful for numerous other investigations and questions of a variety of interests. In this manuscript, the authors use this dataset and compare it to data they previously generated for mouse retinal development to identify 2 new regulators of cone generation and shed insights into their regulation and their integration into the network of regulatory elements within the 13LGS compared to mouse.Weaknesses:(1) The authors chose to omit several cell classes from analyses and visualizations that would have added to their interpretations. In particular, I worry that the omission of 13LGS rods, early RPCs, and early NG from Figures 2C, D, and F is notable and would have added to the understanding of gene expression dynamics. In other words, (a) are these genes of interest unique to late RPCs or maintained from early RPCs, and (b) are rod networks suppressed compared to the mouse?

We were unable to include 13LGS rods in our analysis due to the extremely low number of cells detected prior to P17. Relative expression levels of cone-promoting transcription factors in 13LGS in early RPCs and early NG cells is shown in Fig. 2H. Particularly when compared to mice, we also observe elevated expression of cone-promoting genes in early-stage RPC and/or early NG cells. These include Zic3, Onecut2, Mef2c, and Pou2f1, as well as transcription factors that promote the differentiation of post-mitotic cone precursors, such as Thrb and Rxrg. Contrast this with genes that promote specification and differentiation of both rods and cones, such as Otx2 and Crx, which show similar or even slightly higher expression in mice. Genes such as Casz1, which act in late NG cells to promote rod specification, are indeed downregulated in 13LGS late NG cells relative to mice. We have modified the text to clarify these points, as shown below:

“To further characterize species-specific patterns of gene expression and regulation during postnatal photoreceptor development, we analyzed differential gene expression, chromatin accessibility, and motif enrichment across late-stage primary and neurogenic progenitors, immature photoreceptor precursors, rods, and cones. Due to their very low number before time point P17, we were unable to include 13LGS rods in the analysis.”

“In contrast, two broad patterns of differential expression of cone-promoting transcription factors were observed between mouse and 13LGS.”

“First, transcription factors identified in this network that are known to be required for committed cone precursor differentiation, including Thrb, Rxrg, and Sall3 [25,26,45], consistently showed stronger expression in late-stage RPCs and early-stage primary and/or neurogenic RPCs of 13LGS compared to mice.”

“Second, transcription factors in the network known to promote cone specification in early-stage mouse RPCs, such as Onecut2 and Pou2f1, exhibited enriched expression in early and latestage primary and/or neurogenic RPCs of 13LGS, implying a heterochronic expansion of conepromoting factors into later developmental stages.”

“In contrast, genes such as Casz1, which act in late neurogenic RPCs to promote rod specification, are downregulated in 13LGS late neurogenic RPCs relative to mice.”

(2) The authors claim that the majority of cones are generated by late RPCs and that this is driven primarily by the enriched enhancer network around cone-promoting genes. With the temporal scRNA/ATACseq data at their disposal, the authors should compare early vs late born cones and RPCs to determine whether the same enhancers and genes are hyperactivated in early RPCs as well as in the 13LGS. This analysis will answer the important question of whether the enhancers activated/evolved to promote all cones, or are only and specifically activated within late RPCs to drive cone genesis at the expense of rods.

This is an excellent question. We have addressed this question by analyzing both expression of the cone-promoting genes identified in C2 and C3 in Figure 2C and accessibility of their associated enhancer sequences, which are shown in Figure 6B, in early and late-stage RPCs and cone precursors. The results are shown in Author response image 1 below. We observe that cone-promoting genes consistently show higher expression in both late-stage RPCs and cones. We do not observe any clear differences in the accessibility of the associated enhancer regions, as determined by snATAC-Seq. However, since we have not performed CUT&RUN analysis in embryonic retina for H3K27Ac or any other marker of active enhancer elements, we cannot determine whether the total number of active enhancers differs between early and late-stage RPCs. We suspect, however, this is likely to be the case, given the differences in the expression levels of these genes.

**Author response image 1. sa3fig1:** Relative expression levels of cone-promoting genes and accessibility of enhancer elements associated with these genes in early- and late-stage RPCs and cone precursors.

(3) The authors repeatedly use the term 'evolved' to describe the increased number of local enhancer elements of genes that increase in expression in 13LGS late RPCs and cones. Evolution can act at multiple levels on the genome and its regulation. The authors should consider analysis of sequence level changes between mouse, 13LGS, and other species to test whether the enhancer sequences claimed to be novel in the 13LGS are, in fact, newly evolved sequence/binding sites or if the binding sites are present in mouse but only used in late RPCs of the 13LGS.

Novel enhancer sequences here are defined as having divergent sequences rather than simply divergent activity. This point has been clarified in the text, with the following changes made:

“However, most of these elements mapped to sequences that were not shared between 13LGS and mouse, with intergenic enhancers exhibiting particularly low levels of conservation (Fig. 5B).”

“...demonstrated far greater motif enrichment in active regulatory elements in 13LGS than in mice, though few of these elements mapped to sequences that were shared between 13LGS and mouse (Fig. 5C,D, Table ST10).”

(4) The authors state that 'Enhancer elements in 13LGS are predicted to be directly targeted by a considerably greater number of transcription factors than in mice'. This statement can easily be misread to suggest that all enhancers display this, when in fact, this is only the conepromoting enhancers of late 13LGS RPCs. In a way, this is not surprising since these genes are largely less expressed in mouse vs 13LGS late RPCs, as shown in Figure 2. The manuscript is written to suggest this mechanism of enhancer number is specific to cone production in the 13LGS- it would help prove this point if the authors asked the opposite question and showed that mouse late RPCs do not have similar increased predicted binding of TFs near rodpromoting genes in C7-8.

The Reviewer’s point is well taken, and we agree that this mechanism is unlikely to be specific to cone photoreceptors, since we are simply looking at genes that show higher expression in late-stage neurogenic RPCs in 13LGS. We have changed the relevant text to now state:

“Enhancer elements associated with cone-specific genes in 13LGS are predicted to be directly targeted by a considerably greater number of transcription factors in late-stage neurogenic RPCs than in mice, as might be expected, given the higher expression levels of these genes.”

**Recommendations for the authors:**

**Reviewer #1 (Recommendations for the authors):**
(1) Minor: Clusters C1-C8 (Figure 2) are labeled as "C1-8" in the text but "G1-8" in the figure.

This has been done.

(2) Minor: Showing other neurogenic factors (Olig2, Ascl1, Otx2) and late-stage specific factors (Lhx2, Sox8, Nfia/b) could be shown in Figure 2 to better support the text.

This has been done. These motifs are consistent in both species, but Figure 2F shows differential motifs. The reference to Figure 2F has been altered to include Table ST4, while Neurod1 motifs are shown in Fig. 2F.

**Reviewer #2 (Recommendations for the authors):**
(1) Figure 22A-B: The exclusion of early-stage data from the species-integrated analysis is puzzling, as it could reveal significant differences between early-stage neurogenic progenitors in mice and late-stage progenitors in 13LGS that both give rise to cones. This analysis would also shed light on how cone-promoting transcription factors are suppressed in mouse early-stage progenitors, limiting the window for cone genesis.2C: The figure labels G1-8, while C1-8 are referenced in the text.2F: Neurog2, Olig2, Ascl1, and Neurod1 are mentioned in the text but not labeled in the figure.

2A-B: There are indeed substantial differences between early-stage RPC in 13LGS and latestage RPC in mice that are broadly linked to control of temporal patterning, which are mentioned in the text. For instance, early-stage RPCs in both animals express higher levels of Nr2f1/2, Meis1/2, and Foxp1/4, while late-stage RPCs express higher levels of Nfia/b/x, indicating that core distinction between early- and late-stage RPCs is maintained. What most clearly differs in 13-LGS is the sustained expression of a subset of cone-promoting transcription factors in late-stage RPCs that are normally restricted to early-stage RPCs in mice. However, as mentioned in response to Reviewer #3’s first point, we do observe some evidence for increased expression of cone-promoting transcription factors in early-stage RPCs and NG cells of 13LGS relative to mice, although this is much less dramatic than observed at later stages. We have modified the text to directly mention this point. G1-8 has been corrected to C1-8 in the figure, a reference to Table ST4 has been added in discussion of neurogenic bHLH factors, and Fig. 2F has been modified to label Neurod1.

“First, transcription factors identified in this network that are known to be required for committed cone precursor differentiation, including Thrb, Rxrg, and Sall3 [25,26,45], consistently showed stronger expression in late-stage RPCs and early-stage primary and/or neurogenic RPCs of 13LGS compared to mice.”

“Second, transcription factors in the network known to promote cone specification in early-stage mouse RPCs, such as Onecut2 and Pou2f1, exhibited enriched expression in early and latestage primary and/or neurogenic RPCs of 13LGS, implying a heterochronic expansion of conepromoting factors into later developmental stages.”

(2) Figure 3In 3F, the cone density in the WT retina is approximately 0.25 cones per micron, while in the Zic3 cKO retina, it is about 0.2 cones per micron. However, the WT control in Figure S6C also shows about 0.2 cones per micron, raising questions about whether there is a genuine decrease in cone number or if it results from quantification variability. Additionally, the proportion of cone cells in the Zic3 cKO scRNA-seq data shown in Figure S4E appears comparable to the WT control, which is inconsistent with the conclusion that Zic3 cKO leads to reduced cone production. Therefore, I found that the conclusion that Zic3 is necessary for cone development is not supported by the data.

The cone density counts in the two mutant lines and accompanying littermate controls were collected by blinded counting by two different observers (R.A. for the Zic3 cKO and N.P. for the Mef2c cKO). We believe that the ~20% difference in the observed cone density in the two control samples likely represents investigator-dependent differences. These can exceed 20% between even highly skilled observers when quantifying dissociated cells (PMID: 35198419) and are likely to be even higher for immunohistochemistry samples. Since both controls were done in parallel with littermate mutant samples, we therefore stand by our interpretation of these results.

(3) Figures 4 and 5These figures are duplicates. In Figure 4, Mef2C overexpression in postnatal progenitors leads to increased numbers of neurogenic RPCs, suggesting it may promote cell proliferation rather than inhibit rod cell fate or promote cone cell fate. Electroporation of plasmids into P0 retina typically does not label cone cells, as cones are born prenatally in mice. Given the widespread GFP signal in Figure 4D, the authors should consider that the high background of GFP signal may have misled the quantification of the result.

The figure duplication has been corrected. We respectfully disagree with the Reviewer’s statement that ex vivo electroporation performed at P0, as is the case here, does not label cones. We routinely observe small numbers of electroporated cones when performing this analysis. Cones at this age are located on the scleral face of the retina at this age and therefore in direct contact with the buffer solution containing the plasmid in question (c.f. PMID: 20729845, 31128945, 34788628, 40654906). Furthermore, since the level of GFP expression that is used to gate electroporated cells for isolation using FACS is typically considerably less than that used to identify a GFP-positive cell using standard immunohistochemical techniques, making it difficult to directly compare the efficiency of cone electroporation between these approaches. We agree, however, that Mef2c overexpression seems to broadly delay the differentiation of rod photoreceptors, and have modified the text to include discussion of this point.

“Although a few GFP-positive electroporated cells co-expressing the cone-specific marker Gnat2 were detected in control (likely due to the electroporation of cone precursors, which we have previously observed in P0 retinal explants (Clark et al., 2019; Leavey et al., 2025; Lyu et al., 2021; Onishi et al., 2010)), there was a significant increase in double-positive cells in the test condition, matching the novel cone-like precursor population found in the scRNA-Seq (Fig. 4E).”

“Indeed, overexpression of Mef2c increased the number of both neurogenic RPCs and immature photoreceptor precursors, suggesting that rod differentiation was broadly delayed.”

(4) Figure S2The figure legend lacks information about panels A and B. It is unclear which panels represent immunohistochemistry and which represent RNA hybridization chain reaction. Overall, the staining results are difficult to interpret, as it appears that all examined RNAs/proteins are positively stained across the sections with varying background levels. Specificity is hard to assess. For instance, in Figure S2B, the background intensity of Zic3 staining varies inconsistently from P1 to P24. The number of Zic3 mRNA dots seems to peak at P5 and decrease at P10, which contradicts the scRNA-seq results showing peak expression in mature cones.

The figure legend has been corrected. Negative controls are now included for both in situ hybridization (Fig. S2C’) and immunostaining (Fig. S2G) at P24, along with paired experimental data. We have quantified the total fraction of Otx2+ cells that also contain Zic3 foci, and find that coexpression peaks at P5 and P10. This is now included as Fig. S2E.

The number of Zic3 foci is in fact higher at P5 than P10, with XX foci/Otx2+ cell at P5 vs. YY foci/Otx2+ cell at P10.

“Fluorescent in situ hybridization showing co-expression of (A) Pou2f1 and Otx2 or (B) Zic3, Rxrg, and Otx2 in P1, P5, P10, and P24 retinas. Insets show higher power images of highlighted areas. (C) Zic3, Rxrg, and Otx2 fluorescent in situ hybridization from P24 with matched (C’) negative controls. (D) Pou2f1 and Otx2 fluorescent in situ hybridization from P24 with matched (D’) negative controls. (E) Quantification of the fraction of Otx2-positive cells in the outer neuroblastic layer (P1, P5) and ONL (P10, P24) that also express Zic3. (F) Immunohistochemical analysis Mef2c and Otx2 expression in P1, P5, P10, and P24 retinas. (G) Mef2c and Otx2 immunohistochemistry from P24 with matched (G’) negative controls. Negative controls for fluorescent in situ hybridization omit the probe and for immunohistochemistry omit primary antibodies. Scale bars, 10 µm (S2A-F), 50 µm (S2G) and 5 µm (inset). Cell counts in E were analyzed using one-way ANOVA analysis with Sidak multiple comparisons test and 95% confidence interval. ** = p <0.01, **** = p <0.0001, and ns = non-significant. N=3 independent experiments.”

(5) Figure S3In S3A and S3B, the UMAPs of the empty vector-treated groups are distinctly different. The same goes for Zic3+Pou2F1 UMAPS.In S3A, Zic3 overexpression alone does not appear to have any impact on cell fate. It is not evident that Zic3, even in combination with Pou2F1, has any significant impact on cone or other cell type production, as the proportions of the cones and cone precursors seem similar across different groups.In S3B, Zic3+Pou2F1 seems to increase HC-like precursors without increasing cone-like procursors or cones.Moreover, the cone-like precursors described do not seem to contribute to cone generation, as there is no increase in cones in the adult mouse retina; rather, these cells resemble rod-cone mosaic cells with expression of both rod- and cone-specific genes.

As the Reviewer states, we observe some differences in the proportion of cell types in both control and experimental conditions between the two experiments. Notably, relatively more photoreceptors and correspondingly fewer progenitors, bipolar, and amacrine cells are observed in the samples shown in Fig. S3A relative to Fig. S3B. However, these represent two independent experiments. Cell type proportions seen across independent ex vivo electroporation experiments such as these can be affected by a number of variables, including precise developmental age of the samples, electroporation efficiency, cell dissociation conditions, and ex vivo growth conditions. Some differences are inevitable, which is why paired negative controls must always be done for results to be interpretable.

In both experiments, we observe that overexpression of Zic3, Pou2f1, and most notably Zic3 and Pou2f1 lead to an increase in the relative fraction of cone-like precursors. In the experiment shown in Fig. S3B, we also observe that Zic3 alone, Onecut1 alone, and Zic3 and Pou2f1 in combination also promote generation of horizontal-like cells. All treatments likewise induce expression of different subsets of cone-enriched genes in the cone-like precursors, while also suppressing rod-specific genes in these same cells.

Total numbers and relative fractions of each cell type are now included in Table ST5.

(6) Figure S4The proportion of cone cells in the Zic3 cKO scRNA-seq data shown in Figure S4E appears comparable to the WT control, contradicting the conclusion that Zic3 cKO leads to reduced cone production.

Total numbers and relative fractions of each cell type are now included in Table ST6.

(7) Figure S5In Figure S5A, Mef2C overexpression does not decrease expression of the rod gene Nrl.

This is correct, and is mentioned in the text.

“No obvious reduction in the relative number of Nrl-positive cells was observed (Fig. S5A).”

**Reviewer #3 (Recommendations for the authors):**
(1) The authors make several broad and definitive statements that have the potential to confuse readers. In the first sections of Results: 'retinal ganglion cells and amacrine cells were generated predominantly by early stage progenitors' but later say 'late-stage RPCs in 13LGS retina are competent to generate cone photoreceptors but not other early born cell types.' In the discussion, the authors themselves point out limitations of analyses without birthdating. These definitive statements should be qualified/amended.

Both single-cell RNA and ATAC-Seq analysis can be used to accurately profile cells that have recently exited mitosis and committed to a specific cell fate. When applied to data obtained from a developmental timecourse such as is the case here, this can in turn serve as a reasonable proxy for generating birthdating data. Nonetheless, we have modified the text to state that BrdU/EdU labeling is indeed the gold standard for drawing conclusions about cell birthdates, and should be used to confirm these findings in future studies.

“The expected temporal patterns of neurogenesis were observed in both species: retinal ganglion cells and amacrine cells were generated predominantly in the early stage, whereas bipolar cells and Müller glia were produced in the late stage.”

“Though BrdU/EdU labeling would be required to unambiguously demonstrate species-specific differences in birthdating, our findings strongly indicate that 13LGS exhibit a selective expansion of the temporal window of cone generation, extending into late stages of neurogenesis.”

This sentence does not make a definitive statement about 13LGS RPC competence, and we have left it unaltered.

“These findings suggest that late-stage RPCs in 13LGS retina are competent to generate cone photoreceptors but not other early-born cell types…”

(2) Figure 2C clusters are referred to as C1-8 in the text but G1-8 in the figure. This is confusing and should be fixed.

This has been corrected.

(3) The authors refer to many genes that show differential expression in Figure 2F, but virtually none of these are labelled in the heatmap, making it hard to follow the narrative.

Figure 2F represents transcription factor binding motifs that are differentially active between mouse and 13LGS, not gene expression. We have modified the figure to include names of all differentially active motifs discussed in the text, and otherwise refer the reader to Table ST4, which includes a list of all differentially expressed genes.